# Machine learning-aided design and screening of an emergent protein function in synthetic cells

Shunshi Kohyama [1,2], Béla P. Frohn [1,2], Leon Babl[1] & Petra Schwille [1]

Recently, utilization of Machine Learning (ML) has led to astonishing progress in computational protein design, bringing into reach the targeted engineering of proteins for industrial and biomedical applications. However, the design of proteins for emergent functions of core relevance to cells, such as the ability to spatiotemporally self-organize and thereby structure the cellular space, is still extremely challenging. While on the generative side conditional generative models and multi-state design are on the rise, for emergent functions there is a lack of tailored screening methods as typically needed in a protein design project, both computational and experimental. Here we describe a proof-of-principle of how such screening, in silico and in vitro, can be achieved for ML-generated variants of a protein that forms intracellular spatiotemporal patterns. For computational screening we use a structure-based divide-and-conquer approach to find the most promising candidates, while for the subsequent in vitro screening we use synthetic cell-mimics as established by Bottom-Up Synthetic Biology. We then show that the best screened candidate can indeed completely substitute the wildtype gene in *Escherichia coli*. These results raise great hopes for the next level of synthetic biology, where ML-designed synthetic proteins will be used to engineer cellular functions.

The design of artificial proteins to perform specific functional tasks is one of the ultimate goals of synthetic biology, with the hope to utilize proteins as nano-machines in vivo in cells as well as in vitro in industrial applications[1–3]. In the last two years, the introduction of Machine Learning (ML) based generative models has yielded major breakthroughs in protein design and engineering[2–8]. These methods have led to great advances in the generation of proteins with individual functionality, that is, functionality that depends on the protein alone. Examples are catalytic activity[5–8], small molecule binding[5,6], or spike protein capping[5]. However, the design of proteins with emergent, or higher-order functions, that is, complex functionality that may only be observed indirectly when embedded in a specific biological system, such as biological pattern formation or membrane deformation, is still in its infancy. Proteins that are involved in such functions often exhibit switch-like conformational states and finely tuned cooperative interactions with other proteins, lipids, or nucleotides, which is still challenging to computationally design[9] and predict[10–12]. Importantly, many large-scale intracellular processes of core relevance to life, such as cell migration or division, depend on emergent functions the design and engineering of which would be of great interest for both fundamental and applied research.

Progress towards the design of proteins with emergent function faces two major hurdles, which correspond to the typical workflow of protein design projects: First, sequences are computationally generated, and second, they are both computationally and experimentally screened. For the design of an emergent function, the generation of customized amino acid sequences is a nontrivial task, as the desired function often results from a combination of several simpler subfunctions. Initial steps in this direction are taken by the rise of conditional generative models that are for example conditioned on Gene

[1]Dept. Cellular and Molecular Biophysics, Max Planck Institute of Biochemistry, Martinsried D-82152, Germany. [2]These authors contributed equally: Shunshi Kohyama, Béla P. Frohn. ✉e-mail: schwille@biochem.mpg.de

Ontology terms[13] or Enzyme Condition Numbers[8,14], as well as diffusion models that can take into account non-protein atoms such as DNA or small molecules[15]. Such methods promise to advance protein design from generating hyper-stable, "rock-like" proteins towards the sampling of sequences with the necessary conformational flexibility to give rise to complex functions[16]. For the screening step, a very large number of sequences that are typically generated in the design phase is then filtered first computationally (in silico) and second experimentally (in vitro), often resulting in multiple design-and-screen iterations[2]. Hence, for the screening of proteins designed for higher-order functions, adequate screening methods both in silico and in vitro must be established.

On the computational side, prediction of protein function is still an enormously difficult problem, despite progress being made again by utilization of ML[3,10,17]. Importantly, compared to the prediction of protein affinities, where models perform decently already, higher-order biological functions are much harder to predict[10,11]. On the experimental side, proteins often need specific cellular environments to unfold emergent functions, such as spatial confinement and/or specific membrane composition and geometry, and they often show unexpected behaviors when observed in simplified in vitro environments[18]. For such specialized functional requirements, there exists no established in vitro screening system to date.

Here, we present a proof-of-principle of how ML-generated proteins can be screened for emergent function by an efficient combination of in silico and in vitro screening. Specifically, we developed a computational and experimental pipeline to screen ML-generated variants of the bacterial MinDE system for biological pattern formation. In *Escherichia coli*, the two proteins MinD and MinE engage in ATP-driven reaction-diffusion dynamics that result in membrane-concentrated protein oscillations between the cell poles, placing the division ring at mid-cell and thus determining the division site (Supplementary Fig. 1). Since these oscillations can be reconstituted in vitro within closed lipid compartments[19,20] and on the lipid membrane as a matrix[21–23], and depend on complex interactions between proteins, lipids and ATP[24], the MinDE system is a widely used model of a biological system with higher-order function. Therefore, it is an ideal test system to develop a proof-of-principle screening pipeline for emergent functions.

As conditional models to entirely de novo generate proteins with emergent functions are still in their infancy and mostly not experimentally validated, and since we here we are focusing on the development of a screening pipeline, we utilize an established evolution-based ML model, the MSA-VAE[25], to generate variants of the MinE protein. This method generates a diverse set of proteins where functionality is expected to vary between variants, hence providing an ideal test-set for our screening approach. We then describe and validate screening processes to efficiently assess these variants both computationally and experimentally. For the computational screening, we use a divide-and-conquer approach to score how likely variants are to give rise to the higher-order function based on individual sub-functions. As the MinDE system is well studied, such individual sub-functions of MinE are known that are necessary to give rise to the functional emergence. Hence, we predict and score these functions, namely dimerization, membrane binding, and protein complex formation with MinD, the non-trivial combination of which result in the desired emergent function of pattern formation. Importantly, we show that such a "divide-and-conquer" approach outperforms traditional function-estimation approaches based on sequence similarity or HMM profiles. While here we developed a specialized screening method for the MinDE system, it will be easy to adapt this approach for other higher-order protein functions in the future, based on theoretical or experimental knowledge of complex systems and protein behavior. For the experimental screening of cell-level spatiotemporal patterns, we utilize lipid droplet-based synthetic cell models as utilized by

bottom-up synthetic biology[19,20]. These minimal systems provide a highly customizable, highly controllable environment to characterize proteins particularly by light microscopy-based techniques, and will hence be easy to adapt for other emergent protein functions. Furthermore, we show that cell-free protein expression systems, which have recently become a vital technique in synthetic biology as they can deliver various peptide/protein libraries[26] for prototyping in a rapid and easy manner, can significantly speed up the experimental screening process.

Importantly, we demonstrate that the variant that we have identified as performing best in the in silico and in vitro screening pipeline can fully functionally substitute the wild-type MinE gene in *E. coli* cells. This shows the great potential of divide-and-conquer in silico and synthetic-cell-based in vitro screening for the design of proteins with complex emergent functions. We propose that this and similar pipelines following a combined in silico, in vitro, and in vivo (i[3]) screening approach (Fig. 1) will open the door to the next level of protein design, where the combination of computational design and experimental screening will allow to engineer cellular functions.

## Results

### Generation of artificial MinE homologs

Since non-generative-model-based methods such as random mutagenesis of existing sequences arguably result in mostly nonfunctional proteins, we used a Multiple Sequence Alignment based Variational Autoencoder (MSA-VAE) as introduced by ref. 25. (Fig. 2a) to generate MinE variants. We chose this architecture as it is one of the few methods that is experimentally validated and was shown to have a high success rate[25], thereby outperforming simpler sampling methods based on Hidden Markov Models[25]. On the other hand, we did not utilize de novo protein design models, as we mainly focused on a screening pipeline, but not on design, and the development of a de novo design model capable of the necessary multi-state design would have been a project on its own. The MSA-VAE generates a diverse set of MinE-like proteins with varying functionality expected[25], hence providing an ideal test-set for the development of a screening pipeline. There are relatively few naturally occurring MinE sequences to train the model on, compared to other studies where homologs were generated (~6000 sequences in our dataset, compared to ~17,000 sequences used to train ProteinGAN[27]), so we preferred the MSA-VAE over a GAN as it needs fewer parameters because information about fold and function is already encoded in the MSA[28,29] (ProteinGAN[27] has ~60,000,000 trainable parameters while our model has ~1,000,000). We trained the MSA-VAE with a modified ELBO loss function similar to ref. 25. with a range of different hyperparameters (see Methods) and evaluated performance by single and pairwise amino acid frequency distributions as in the original paper (Supplementary Fig. 2a). A high correlation in this metric indicates that evolutionary constraints are considered when generating sequences[25]. With the selected set of hyperparameters, we generated 4000 variants by passing random draws from a normal distribution through the decoder and using the maximum value at each MSA position to determine amino acids. As can be seen in Fig. 2b, sequence conservation among the generated variants is highly similar to sequence conservation in naturally occurring variants, indicating that the model had generated reasonable sequences, taking into account evolutionary constraints. Dimensionality reduction on the latent space by Principal Component Analysis (Supplementary Fig. 2b) further showed clustering by phylogenetic groups, confirming that the latent space conserved information about sequence relationships. However, some overlaps in the clusters can be observed and the correlation between pairwise amino acid frequencies of natural and generated sequences is not perfect, indicating that the generative model might also introduce some mutations that could impair the function of the protein. Thus, we had generated a set of artificial homologs where different grades of

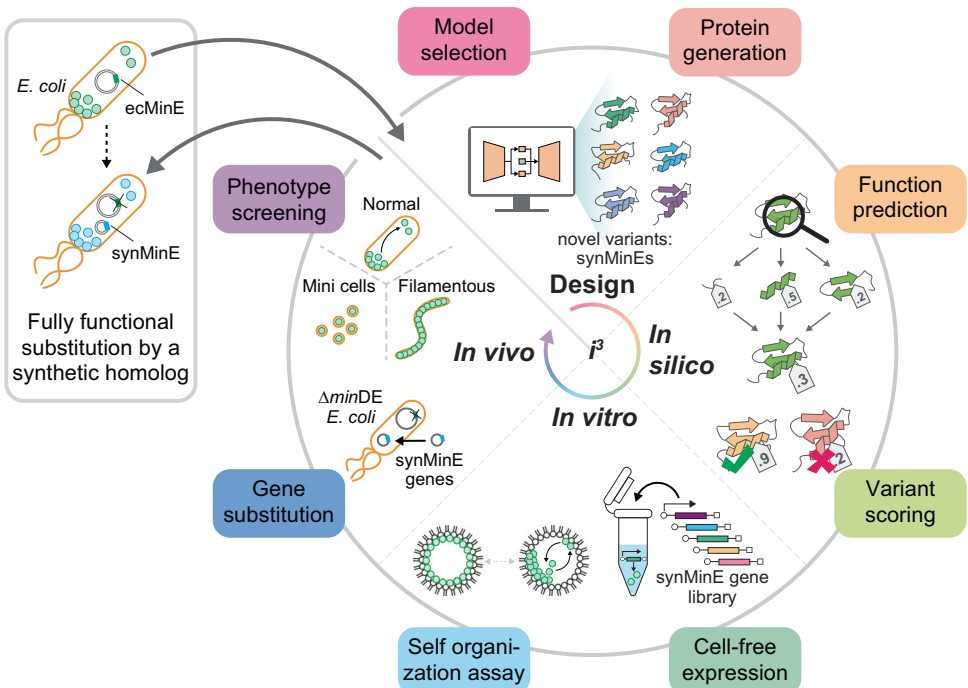

**Fig. 1 | Overview of machine-learning assisted protein design and i³-screening pipeline for artificial homologs.** The combined pipeline of this study: (1) Variational Auto Encoder-based sequence generation, (2) divide and conquer in silico scoring based on structures predicted by AlphaFold2, (3) in vitro cell-free expression and encapsulation in cell mimics testing for self-organization, and (4) in vivo substitution of synMinE genes in Δ*min*DE *E. coli* cells. This pipeline finds an artificial, fully functional homolog of the wild-type protein.

functional performance were expected, which we could then subsequently screen for emergent function.

## In silico scoring of emergent function

We reduced the number of candidate proteins from 4000 to 48 for subsequent experimental in vitro analysis by first screening them computationally. To provide sufficient heterogeneity of this subset, we initially screened proteins based on sequence identity. First, we filtered out all proteins with more than 60% sequence identity to the wild-type MinE in *E. coli*, the organism eventually targeted for in vivo screening. Second, we clustered the remaining generated variants by 60% sequence identity. Third, we randomly selected one sequence per cluster for further analysis. As a result, we got 167 remaining sequences to evaluate in our in silico pipeline.

Here, we defined emergent, or higher-order protein functions as behaviors of a protein that can be measured only indirectly, e.g., via large-scale effects. Such complex functions arise when proteins or protein modules with specific, more easily measurable sub-functions interact to form a system that shows higher-order behavior beyond the sum of the individual functions. Due to the distinct nature of this approach and the sparsity of respective data, qualitative and quantitative predictions of emergent functions are still beyond the scope of ML. Thus, to screen computationally for the potential of our protein variants to show emergent behavior, we hypothesized that a combined screening for the necessary sub-functions could provide an indirect measure of higher-order functions. We call this a "divide-and-conquer approach". In the case of MinE, its higher-order function, volume oscillations through spatiotemporal self-organization with MinD, is known to be composed of three sub-functions[22–24]: (i) membrane binding, (ii) formation of the MinDE complex that stimulates MinD's ATPase activity, and (iii) homo-dimerization (Fig. 2c upper panel). To evaluate the expected functionality of the generated variants, we first predicted their structures using AlphaFold2 Multimer[30], and then developed an in silico pipeline to estimate the three sub-functions from the structure. Thus, we used a full sequence-structure-function

pipeline (Fig. 2a). Interaction to MinD and homodimerization was evaluated based on the Predicted Align Error (PAE) of the AlphaFold2 Multimer[30] output, similar to other protein design studies[4,5]. The membrane binding capability was estimated by calculating the hydrophobicity of the N-terminal region using ProteinSol Patches[31], since the hydrophobic interaction between the N or C-terminal region of proteins and lipid molecules is a determinant factor of the membrane binding[32–34] (see Methods and Supplementary Fig. 2c). As we eventually wanted to test the proteins in *E. coli* cells, we also predicted solubility of the proteins in *E. coli* using ProteinSol[35] as fourth score (Supplementary Fig. 2c). All four scores were normalized and summed up, resulting in a roughly normally distributed final Function Score (Fig. 2a lower panel). While here we weighted each score equally, in future similar studies each sub-function could be weighted by importance to the emergent function, if such information is available (see below for post-hoc analysis of the scores used here). We then sorted the 167 heterogeneous variants by this score and validated the ranking visually. As can be seen representatively in Fig. 2c, proteins with low scores tend to be predicted to miss a proper interaction interface with MinD and to have a disordered and either very long or very short N-terminal region, suggesting impaired MinD's ATPase stimulation and membrane binding. Proteins with high scores tend to resemble the wild type closely. Interestingly, it is known that a conformational change is needed to swap from MinE-MinE homodimer to MinE-MinD heterodimer[24], and among the low-scoring variants such a change was often not predicted (Supplementary Figs. 3, 4). We then chose and double-blinded the best-scoring and worst-scoring 24 sequences for experimental screening (Fig. 2a lower panel, Supplementary Fig. 3, 4, and Supplementary Data 1), in order to validate our in silico scoring approach.

## In vitro screening for emergent function using a cell-free system

The first step in the experimental screening of newly designed proteins is typically the expression of target proteins in *E. coli* cells followed by chromatographic purification protocols[6,7,25,27]. However,

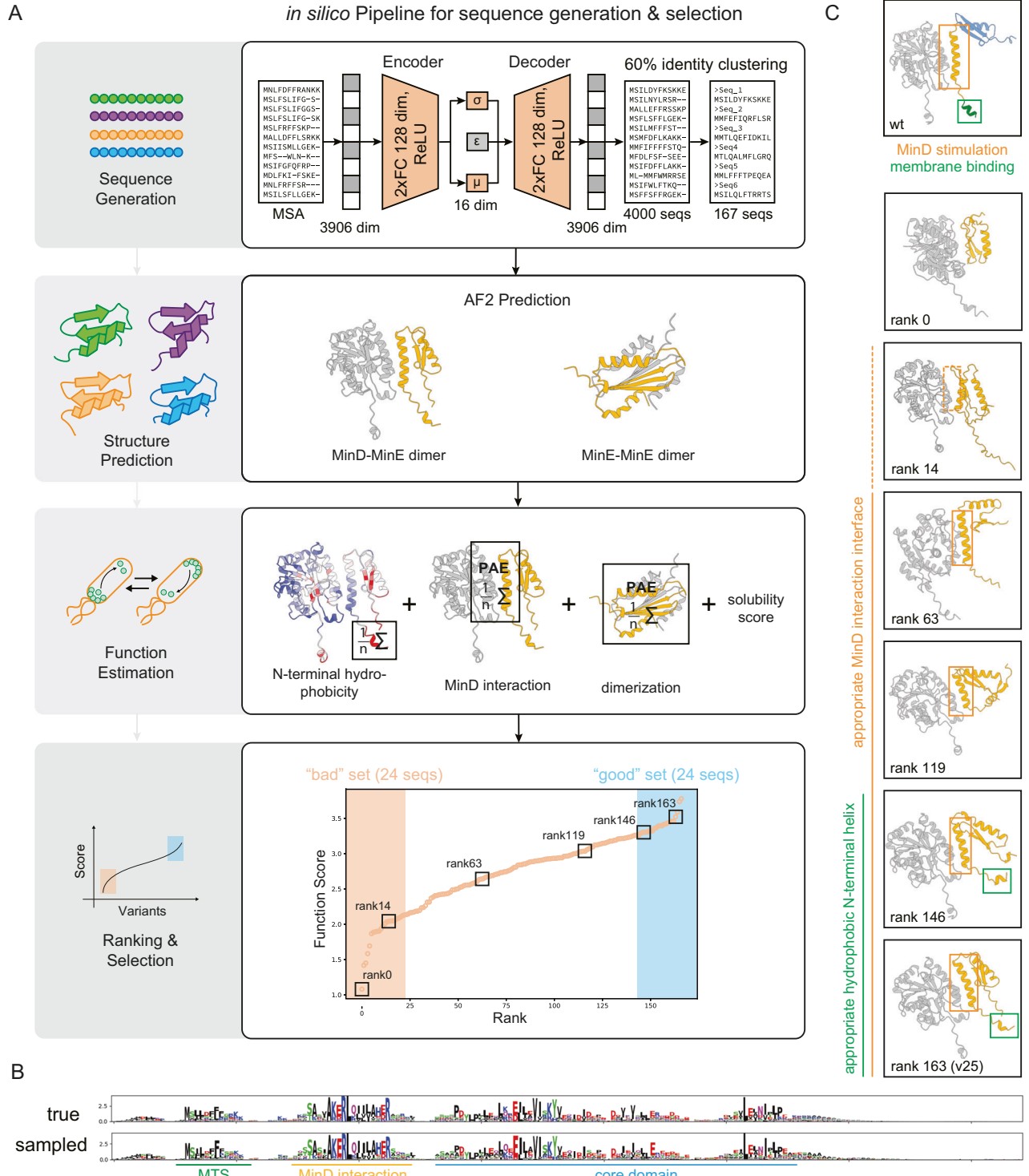

**Fig. 2 | Generation of MinE homologs and in silico screening for expected function. a** Pipeline overview. Sequences are generated using a Variational Autoencoder and clustered by 60% identity to ensure heterogeneity. The structures of the remaining 167 sequences are predicted using AlphaFold2 for homo- and heterodimers. A function score is calculated based on solubility and the three subfunctions known to allow MinE to oscillate in *E. coli*: N-terminal membrane binding, interaction with MinD, and dimerization. The results are ranked, and the best and worst 24 candidates are selected for in vitro analysis. **b** Sequence conservation in naturally occurring and newly generated MinE homologs are highly similar. **c** Visual validation of ranked structures. With better ranking, structures show better MinD-interaction sites and membrane targeting regions.

this approach comes with many difficulties in experimental optimization due to protein solubility, cell toxicity, etc. Instead, to accelerate the screening pipeline and make it more generally applicable, we utilize an in vitro cell-free protein synthesis system[36,37], where target proteins can typically be expressed within 1 h of incubation of

a mixture of transcription-translational factors and DNA/mRNA templates encoding the target proteins. The transcription-translational factors can be typically obtained from either a lab-made cell lysate or a commercially available cell-free protein synthesis kit. Such cell-free expression systems have an enormous

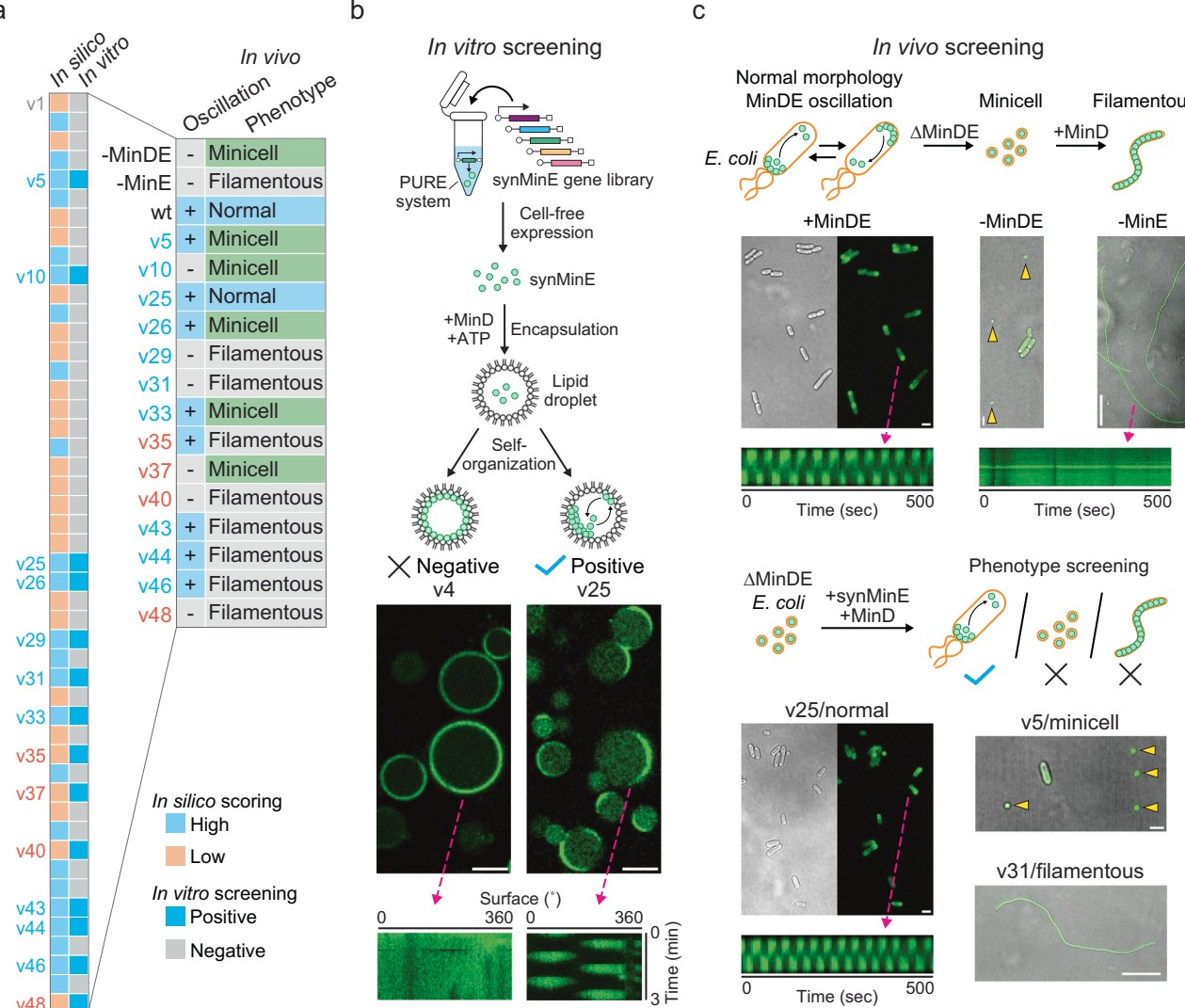

**Fig. 3 | In vitro and in vivo screening for emergent functions of synMinE variants.** **a** Summary of the i³-screening results for functional synMinE variants. The colors of each variant name indicate whether they are scored high (blue) or low (orange) by in silico screening. **b** In vitro screening of synMinE variants. Proteins were synthesized in the PURE cell-free expression system from a gene library of synMinE variants and then encapsulated in lipid droplets with EGFP-MinD (shown in green) and ATP. Emerging Min oscillation patterns as visualized in the kymograph on the bottom-right panel were observed with 14 variants from 48 candidates and further tested in vivo. Scale bars: 20 µm. **c** In vivo screening of synMinE variants. The remaining 14 variants were introduced in Δ*min*DE *E. coli* cells with GFP-tagged

MinD. Subsequently, cell morphology and Min oscillation were validated to identify the functional synMinE variants. synMinEv25 fully substituted the wild type (+MinDE condition). Differential interference contrast and fluorescence images are separately indicated for wild type (+MinDE) and synMinEv25 (v25), or merged for the other conditions shown in the figure. Kymographs show Min oscillations inside the cells with wild type or synMinEv25, while MinD did not induce oscillations without MinE (-MinE). Yellow arrows indicate the minicells in -MinDE and v5 conditions. Scale bars: 2 µm (+MinDE, −MinDE, v25, and v5) or 20 µm (−MinE and v31). All the micrographs correspond to a reproducible result from 3 or more independent biological replicates.

potential to be further utilized in various experimental setups. For instance, different protein synthesis systems are available which can easily be adapted to the target/original organism, protein size, required protein chaperons, or expected expression yield. In this study, we tailored our designed proteins for in vivo function in *E. coli* and thus chose the *E. coli*-based cell-free synthesis platform called PURE system[38]. It has been previously demonstrated that the PURE system can efficiently synthesize functional Min proteins, which self-organize into dynamic wave patterns in vitro in cell-mimicking environments such as lipid containers[39–42].

We performed in vitro screening of the 48 MinE variants, named synMinEv1–48, by validating whether cell-free expressed synMinE variants could form Min waves in an in vitro reconstitution setup. First, all 48 synMinE variants were synthesized with the PURE system, where

more than 80% (40 variants) of synMinE proteins were expressed at detectable level (Supplementary Fig. 5). Subsequently, each expressed variant was encapsulated within lipid droplets composed of POPC/PG mixture with purified MinD and ATP as cofactors to provide the lipid interaction partner and geometrical constrains needed for oscillation. After checking for Min waves by laser scanning microscopy, we found in total 14 positive variants that give rise to spatiotemporal patterns on the lipid membrane with typical Min wave patterns (traveling wave and pole-to-pole oscillation), as well as typical oscillation periods (1–2 min) in lipid droplets as previously reported[39] (Fig. 3a, b, Supplementary Fig. 6, and Supplementary Movie 1). The other 34 variants did not show any heterogeneous localization over the usual time scale of Min wave emergence (5–15 min) (Supplementary Fig. 6 and Supplementary Movie 2).

Reassuringly, after unblinding the variants, we found that 10 of these positive variants are from the high in silico scoring candidates and 4 variants are from the low in silico scoring candidates. Post-hoc analysis (Supplementary Fig. 7) showed that the function score we had assigned during the in silico screening indeed significantly distinguished between variants that showed oscillations in vitro vs. variants that did not (Mann−Whitney−Wilcoxon test: $p = 0.03$, AUC = 0.68). We also found that the function score only minimally correlates with sequence similarity/identity to natural MinEs (Supplementary Fig. 8). Interestingly, post-hoc we furthermore found that two of the four sub-function scores, solubility and dimerization, could not significantly distinguish positive from negative variants (Mann-Whitney-Wilcoxon test: $p_{solubility} = 0.79$, $p_{dimerization} = 0.22$). Inspired by this we generated an improved function score by only combining the scores for MinD interaction and N-terminal hydrophobicity. Astonishingly, this improved score could not only almost perfectly distinguish between positive and negative variants (Supplementary Fig. 7, Mann−Whitney−Wilcoxon test: $p = 2e{-}7$, AUC = 0.92), but also outperforms measures typically used for automated function annotation such as sequence similarity to the closest homolog or HMM-profile-based scoring (Supplementary Fig. 7), suggesting that "divide-and-conquer" approaches might be promising for automated labeling of sequence databases. Taken together, these results suggest that in silico screening for higher-order functions is indeed possible by an indirect scoring based on the sub-functions necessary for the emergent behavior. This could be extended to other emergent functions that are composed of measurable sub-functions of one or more proteins, and should also be possible with entirely de novo designed proteins. In addition, the combination of cell-free expression (here we took 1 h for incubation) and quick encapsulation and visualization (15 min/sample) steps enabled us to screen all synMinE variants in 2 days (24 variants/day), showing that such complex function can be efficiently screened for in vitro, as compared to conventional purification procedures.

### In vivo substitution finds a fully functional complement of the wild-type gene

To further investigate whether these bottom-up constructed in vitro systems can truly screen for physiological function in vivo, we then assessed whether those 14 positive variants could also give rise to Min oscillations in *E. coli* cells. The 14 positive synMinE variants were introduced in an *E. coli* strain (HL1) lacking *min*DE genes by transforming plasmids encoding the respective synMinE variant and GFP-tagged MinD, as shown in previous studies[23,43]. With this setup, there are three possible phenotypes[44] (Fig. 3c and Supplementary Fig. 9). First, the normal phenotype, where both MinD and MinE are functional. Second, the minicell phenotype, observed when Min proteins are dysfunctional in division ring placement, i.e., where the division ring is not positioned at mid-cell but at a random position. This results in a certain number of cells (29% of the population in Δ*min*DE control vs 2.1% in the normal phenotype) becoming non-chromosome miniature-sized spherical cells. Third, the filamentous phenotype, observed when Min proteins are dysfunctional in division ring assembly, where MinD occupies the entire inner membrane area and prevents the formation of a division ring at all. Notably, since Min wave dynamics can be observed with some MinE mutants regardless of cell morphology[45,46], synMinE variants may also be able to induce the wave dynamics even in minicell or filamentous phenotype.

Strikingly, we found that 7 out of 10 high in silico scoring synMinE variants evoked Min oscillations inside the cells, while only one of the low-scoring variants showed oscillations (Fig. 3a, c, Supplementary Fig. 9, and Supplementary Movie 3–6). This suggests that the essential requirements for Min wave oscillation might be stricter in vivo than in vitro, potentially because proteins are constricted in even smaller microscopic spaces and other cellular molecules, such as proteins,

DNA, and RNA, could potentially induce non-specific interactions with the Min proteins. In addition, we confirmed that none of the top-5 in silico but in vitro negative variants induced oscillations in vivo (Supplementary Fig. 10), showing that in vitro screening successfully filtered out the non-functional variants. Furthermore, analysis of cell morphology revealed that the majority of wave-inducing synMinE variants, and especially all low-scoring variants, induce the minicell or filamentous phenotype, as a result of dysfunction in division ring assembly or placement (Fig. 3a, c, and Supplementary Fig. 9). This suggests that further complex molecular dynamics of Min proteins, as the interaction of MinD with other proteins competing with MinE, such as the third Min protein, MinC[44,47,48], are essential to position the division machinery at the proper region. Finally, we found that one variant, synMinEv25, fully restores the normal cell phenotype together with Min oscillations (Fig. 3a, c, Fig. 4, Supplementary Movie 3, and 4), representing, to our knowledge, the first functional substitution of a natural gene by an artificial homolog generated by a generative model in a living organism. Intriguingly, synMinEv25 had already outperformed all the other variants in the in vitro wave occurrence scores (Supplementary Fig. 6b), as well as ranked as best candidate in the improved function score in silico (Supplementary Fig. 7), confirming that in vitro scoring as well as in silico scoring could reasonably estimate the emergent protein function, which will considerably enhance the efficiency of experimental validation for emergent functions in coming studies following a similar pipeline.

### Functional analysis of synMinEv25 reveals its impeccable capability

To further understand the function of synMinEv25, we conducted in vivo and in vitro characterization of synMinE variants. First, we analyzed cell growth with all high-scoring synMinE variants. In contrast to the control (-MinE) condition, the introduction of synMinEv25 successfully restored growth rates to the wild-type level (Fig. 4a, Supplementary Fig. 11a, and b). Also, 6 of the 10 positive high-scoring variants restored cell growth as well (Supplementary Fig. 11a), suggesting that even without proper positioning of the division machinery inducing abnormal phenotypes, synMinE variants can induce cell division and growth. We then measured the cell size distribution of normal and minicell phenotype mutants to assess the accuracy of cell division led by Min oscillations. We found that synMinEv25 has a similar minicell population (2.1% (wt) vs 2.3% (v25)), median cell size (3.5 μm (wt) vs 3.4 μm (v25)), and even narrower size distribution than wild type (2.4 μm (wt) vs 1.4 μm (v25) in variance), suggesting synMinEv25 supports proper cell division by placing the division ring in a correct location within comparable temporal and geometrical scales, and especially reduces the population of elongated cells, thus better-conferring functionality in cell division (Fig. 4b). The other variants induced much higher minicell populations (7.2–35%) and wider size distributions (Supplementary Fig. 11c), indicating that they were inefficient in positioning the cell division machinery. The Min oscillations induced by synMinEv25 showed a similar tendency of periods against cell length as the wild type, with slightly slower oscillations (Fig. 4c, Supplementary Fig. 11d, Supplementary Movie 3, and 7). Taken together, synMinEv25 can substitute the wild type in all intrinsic functions of the Min system−cell growth, morphology, and biological pattern formation.

Next, we set out to purify the promising synMinE variants and were able to obtain 6 out of the 10 high-scoring variants, including synMinEv25, in a standard affinity purification protocol (Supplementary Fig. 12a). This success rate of only 60% with already pre-filtered candidates emphasizes the great benefit of cell-free expression for functional screenings, where more than 80% of variants were obtained (Supplementary Fig. 5). We characterized those purified proteins by three functional assays in vitro with regard to the three sub-functions that were tested during the in silico scoring, (i) membrane binding, (ii) catalyzing MinD's ATPase activity, and (iii) oligomerization. Strikingly,

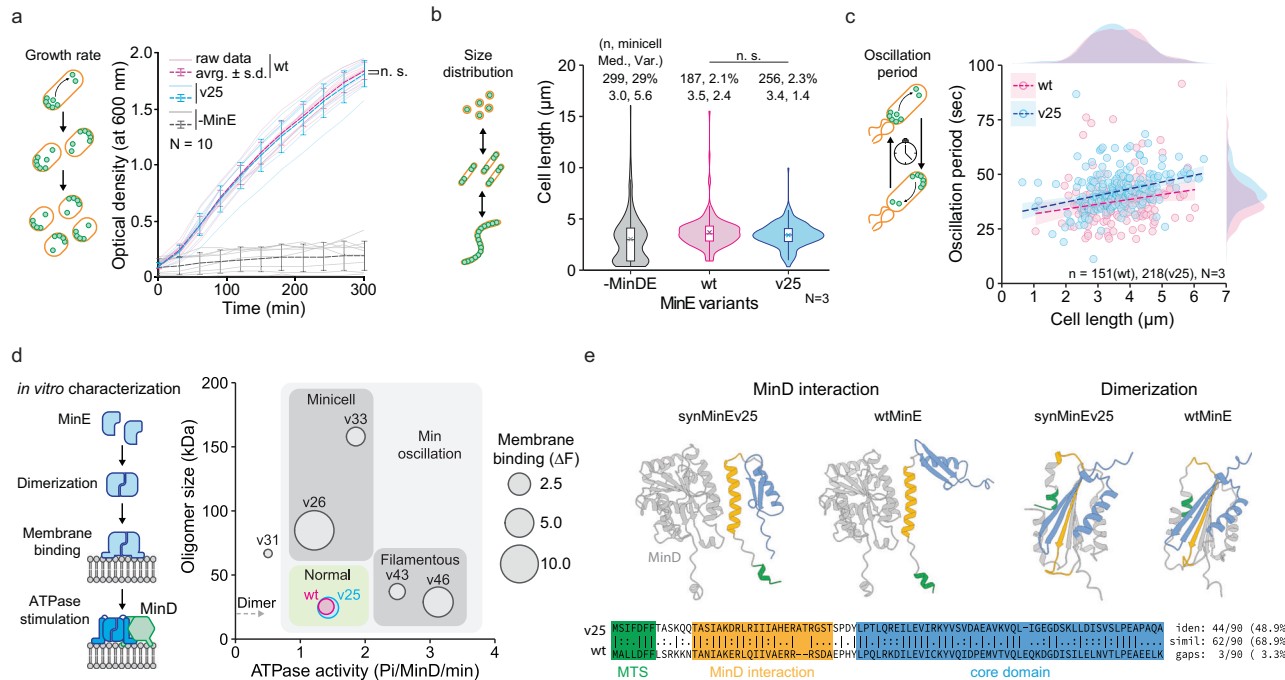

**Fig. 4 | Characterization of synMinEv25 confirms the functional substitution of the wild type. a** Growth curves of *E. coli* cells show that the introduction of syn-MinEv25 in Δ*min*DE cells together with MinD recovers cell growth at the same level as wtMinE (n.s. indicates *p* > 0.05 between wild type (wt) and synMinEv25 (v25) in double-sided Welch's *t* test at 300 min). *N* = 10 biologically independent samples are examined over 10 independent experiments. Error bars indicate average ± standard deviation. **b** Violin plots of cell-size distributions of *E. coli* cells confirm that both wtMinE and synMinEv25 confer proper size distribution while Δ*min*DE (-MinDE) cells produce a high population of minicells (<1 μm in cell length). Box plots inside the violin distribution indicate maximum and minimum in 1.5xIQR, 25th and 75th percentile, median (bar), and mean (cross symbol) values. N.s. indicates *p* > 0.05 in double-sided Mann–Whitney U test. *N* = 3 biologically independent samples are examined over 3 independent experiments. **c** Scatter plots of Min oscillations induced by wtMinE or synMinEv25 exhibit similar period and size dependency, confirming that synMinEv25 can functionally substitute the wild type. The dotted lines and shades indicate linear trends with 95% confidence intervals. Distributions of plots are also shown as external density plots. *N* = 3 biologically independent samples are examined over 3 independent experiments. **d** Bubble plot of in vitro characterization of synMinE variants. synMinEv25 has the closest scores to the wild type among other variants, showing a fine match with the screening results. **e** Comparison between wtMinE and synMinEv25 structures and sequences confirms that synMinEv25 is a proper distant homolog while keeping similar structures to the wild type.

a bubble plot (Fig. 4d, Supplementary Fig. 12b, c, and Supplementary Fig. 13) indicates that synMinEv25 has almost the same scores in all three parameters compared to the wild type, showing that our screening and in vivo characterization are plausible. Moreover, we found an interesting relationship between scores and cell phenotype. Variants with higher ATPase induction activity than the wild type but relatively similar oligomerization scores seem to induce the filamentous phenotype, while variants with ATPase induction comparable to the wild type but bigger oligomer sizes seem to induce the minicell phenotype. This suggests that a delicate balance of those two parameters is particularly important for proper cell division, while the strength of membrane binding seems not to be a determining factor. These facts can be considered as possible weighting factors as mentioned above for in silico scoring to further improve the function estimation in future studies. It should be mentioned, however, that the individual in silico scores and the determined in vitro characteristics showed only diminishing correlations (Supplementary Fig. 14), which is unsurprising given that we only characterized a small number of successful proteins that showed oscillations in vitro.

Finally, we analyzed similarities of our generated variants to natural MinE proteins on the sequence level (Supplementary Fig. 15). Unsurprisingly, residues known to be crucial for MinD ATPase stimulation[46,49] as well as core residues for its conformational changes[46,50–52] are conserved in all functional variants. However, these residues are also conserved in most variants that did not work, and especially also in many of the variants that had low in silico scores. Matching to the minimal correlation of the function score and

sequence identity/similarity to natural MinEs mentioned above (Supplementary Fig. 8), we see this as another sign that our scoring did not simply indirectly test for sequence similarity to wild type variants, but truly for function based on their structures.

Furthermore, sequence comparison showed that the sequence identity of wild type MinE and synMinEv25 is less than 50%, sequence similarity is less than 70% (Fig. 4e), and sequence identity of synMinEv25 and its closest natural homolog is 78.7% (Supplementary Data 1). This validates synMinE clearly as a homolog of existing MinE proteins, where existing similarity-based sequence annotation pipelines would assume a similar function. Interestingly, ML models similar to ours, generating variants of existing proteins, that were recently used to generate versions of enzymes, showed a clear cutoff in experimental validation when the generated sequences go below 80% identity to the respective closest homolog[8,27]. That is, when sequences with lower identity to their closest natural homolog were tested, they showed diminishing catalytic activity. It is quite remarkable that synMinEv25 with 78.7% identity lies at the edge of this empirical cutoff, despite it having not one, as in the case of enzymes, but three sub-functions that are necessary to perform the desired emergent function.

## Discussion

Computational protein design has made impressive advances in the last two years with the introduction of Deep Learning based generative models, and the rise of conditional models, e.g., taking into account Gene Ontology terms[13], and models including non-protein data, like DNA or small molecules[15], brings the computational design of proteins

with emergent functions into reach, that is, emergent functions that may only be observed indirectly when embedded in a specific biological system. The computational design step, however, is only the first in a typical protein design pipeline, which usually needs to be followed by a sophisticated combination of computational and experimental screening procedures. This remains an exceptionally difficult problem, as computational prediction of biological function is still performing poorly compared to the prediction of molecular function[10,11], and high-throughput experimental screening needs to provide the specific environment necessary for the higher-order function. Here, we provide a proof of principle of how such integrated screening, both computationally and experimentally, can be achieved. We generate variants of the bacterial protein MinE, which is part of the pattern-formation system MinDE, often used as a model system in the study of emergent protein functions.

First, we use a divide-and-conquer approach for computational screening, where we predict sub-functions known to be necessary for the emergent function, and sum them up to estimate an overall function score. The pipeline we used here is adapted to predict MinE's sub-functions, namely, membrane binding, MinDE complex formation, and dimerization, and hence not generalizable without modifications. However, the adaptation of the concept to other proteins with emergent function and entirely de novo designed proteins should be straightforward if respective sub-functions can be defined. For example, a similar divide-and-conquer approach has recently been suggested for the de novo design of linear motor proteins, whose higher-order function (linear walking) can be broken down into three sub-functions: track binding, asymmetry, and allosteric control[53]. Similarly, theoretical studies suggest that systems showing biological pattern formation can be dissected to essentially two sub-functions, where one sub-function is for one component of the system to be switchable between two states, whilst the other sub-function is for the other component of the system to facilitate the switching[54]. We argue that protein design and screening informed by such theoretical and experimental dissection of emergent functions into sub-functions will serve as a protein design toolbox, given that accurate methods to predict the necessary sub-functions either already exist or can be developed. It is worth mentioning here, however, that ML-based predictors of protein function are to be treated with a grain of salt, because many such projects still fail to account for the unique biases of protein datasets, such as the fact that protein sequences with similar function mostly evolved from a common ancestor and hence must not be treated as independent samples, which leads to false-high performance metrics if it is not controlled for[55]. We recommend to evaluate function directly based on structural and surface features like we did, as structure predictors are quite trustworthy. In fact, we showed that this approach outperforms sequence-similarity based prediction in the post-hoc analysis.

Second, for experimental screening, we utilize a minimal cell-mimicking in vitro system, as established by bottom-up synthetic biology, that allows a rapid and easy-to-follow pipeline to screen designed proteins for intracellular functions. More specifically, we introduce two core techniques of bottom-up synthetic biology, namely, droplet-based synthetic cell models and cell-free protein expression, to the field of protein design, providing the grounds to efficiently screen for emergent functions of proteins and protein assemblies. Here, we screened for pattern formation inside a membrane compartment. As synthetic cell models are highly modular and customizable, this could easily be adapted to other biological systems that require specific environments for their functions. For example, it would be easy to adapt the pipeline established here to a wide pH range, a variety of salt/ion conditions, crowding environments, or particular types of lipids. Similarly, any kind of interaction partner could easily be included in the encapsulation, as shown here by supplementing purified MinD in the in vitro screening, e.g., DNA or protein

filaments as tracks for novel motors as described above, or DNA condensates to mimic the nucleoid.

On the other hand, by using PURE cell-free protein expression, we were able to speed up the experimental screening process enormously, as there was no need for time-consuming protein purification. We chose the PURE system among different types of cell-free expression because of its easy use and commercial availability, and importantly, an expectation of high expression yield of synMinE proteins, given their small sizes (around 90–100 residues) and simple structures. Indeed, the success rate of PURE cell-free expression of synMinE was more than 80% and was clearly better than 60% success in the purification of those proteins. However, in the case of bigger and difficult-to-fold proteins, it could be done by supplementation of protein chaperones, protein disulfide isomerase, and additional ribosomal factors that are also commercially available, as well as lysate-based cell-free expression systems that are known to provide a comprehensive environment for protein folding and expression efficiency. As both, synthetic cell models and cell-free expression, provide modularity and support different use cases, the screening strategy can be adapted to the specific needs of individual design projects.

Moreover, we showed that a combined in silico and in vitro screening can successfully identify a candidate that not only functions in vitro, but can fully replace the wild type gene in vivo, marking an example of a Machine Learning (ML) generated protein substitution of a natural gene. Remarkably, the MinE homolog from *Neisseria gonorrhoeae*, ngMinE, has a similar sequence identity/similarity to wtMinE as synMinEv25 (ngMinE: 41.6%/71.9%, synMinEv25: 48.9%/68.9%) and can also induce waves in vivo in *E. coli*[56]. However, a significantly lower appearance of waves was reported with ngMinE than we found with MinEv25 (MinEng: ~70% among observed cells, synMinEv25: ~100%), and the oscillation period was about 4 times slower (174 s) than with wtMinE, while synMinEv25 shows a difference of only less than 10% (39 s vs 42 s). Therefore, this natural homolog could not fully functionally substitute the wtMinE in the same way synMinEv25 can. It is easily imaginable how our combination of ML-aided design and in vitro screening could help to not only produce known functions, but to show enhanced or entirely functions in vivo. As a toy problem in the framework of the MinDE system, for example, variants that lead to slower/faster oscillations or different cell length distribution could be screened for, while not affecting the cell viability itself in both cases. Other exciting candidates are cytoskeleton-like proteins that force cells into particular shapes, or motor proteins that facilitate the directed movement of cells are particularly fascinating. Taken together, the proof-of-principle screening system presented here by a combined in silico, in vitro, and in vivo pipeline marks a huge step forward towards customizable biological systems and opens the door to the engineering of whole living organisms from the bottom-up.

## Methods
### Dataset construction
All sequences containing the InterPro[57] domain IPR005527 were downloaded from InterPro on 30/05/2022. They were clustered by cd-hit[58] with 100% identity cutoff, meaning that no redundant sequences were kept. As the dataset was small (8,496 non-identical sequences) and VAEs intrinsically add noise, no further clustering by identity was performed. All sequences shorter than 20 and longer than 200 amino acids and all sequences containing non-standard amino acids were omitted. A Multiple Sequence Alignment (MSA) was calculated using Clustal Omega[59], using the Hidden Markov Model Profile of the MinE domain provided by Pfam[60] (downloaded and extracted on 17/03/2022). To narrow the MSA, columns that contained gaps in over 98% of the sequences were cut out. The remaining dataset consisted of 5,958 sequences and the MSA was 186 columns wide. Finally, sequences were one-hot encoded as input for the VAE and split into a train

(80%) and validation (20%) set. As the final evaluation of the model would be done experimentally, a test split was omitted.

## Variational Autoencoder (VAE)

The Variational Autoencoder followed largely the architecture introduced by Hawinks-Hooker et al.[25]. (Fig. 2a), implemented in PyTorch[61]. We optimized hyperparameters by evaluating the performance of the VAE using the correlation of pairwise amino acid frequencies (see below) of 4000 random samples generated by the VAE with the natural 5,958 sequences, a metric introduced by the original MSA-VAE paper[25]. A high correlation in this metric indicates that evolutionary constraints are considered when generating sequences[25]. In the final model, both Encoder and Decoder consisted of a fully connected neural network with two 128-dimensional hidden layers and ReLU activation function. The latent space was 16 dimensional. After the second hidden layer in the Decoder, a softmax function generates a probability score to observe an amino acid or gap at each position in the MSA. For optimization, the Adam optimizer was used with PyTorch defaults and batch size 8 and learning rate 0.001, and the model was trained for 60 epochs. As loss function a modified version of the ELBO loss was used, where the KL-divergence loss was multiplied with the factor 0.01. During hyperparameter optimization we had found that without such a weighting, the VAE would always generate the same sequence, similar to mode collapse in Generative Adversarial Networks[62]. The final loss function used was

$$\text{Loss} = 0.01\text{Loss}_{\text{KL}} + \text{Loss}_{\text{reconstruction}} = 0.01\text{KL} + \text{BCE} \quad (1)$$

$$= 0.01 \frac{1}{D} \sum_{i=1}^{D} -0.5\left(1 + \sigma_{\log,i} - \mu_i^2 - e^{\sigma_{\log,i}}\right)$$
$$- \frac{1}{L} \sum_{i=1}^{L} y_i \log(\hat{y}_i) + (1 - y_i) \log(1 - \hat{y}_i) \quad (2)$$

where KL is the Kullback–Leibler divergence between the latent distribution and a normal distribution, BCE is binary cross entropy, $D$ is the number of dimensions of the latent space, L is the length of the sequence after one hot encoding and flattening, $\sigma_{\log,i}$ is the logarithmic variance of the $i$th latent dimension, $\mu_i$ is the mean of the $i$th latent dimension, $y_i$ is the true value of the $i$th one hot encoding and $\hat{y}_i$ is the predicted value of the $i$th one hot encoding. $\sigma_{\log,i}$ and $\mu_i$ are the output of the Encoder, $\hat{y}_i$ is the output of the Decoder.

During the Review process, we realized that we had made some unfortunate choices when formulating the loss function, which might have been the cause of the mode collapse behavior because of which we had to introduce the factor 0.01. First, we used a binary cross entropy loss for the reconstruction loss, which treats every amino acid at every position individually. Clearly, it would have been better to use a categorial cross entropy per position, as only one amino acid (or gap) per position should be selected. Second, we normalized the KL-Loss by $D$ ( = 16), whilst we normalized the reconstruction loss by $L = 3906$). Hence, without factor 0.01, we originally weighted the KL loss by a factor of 3906/16 = 244 more than the reconstruction loss. In effect, this means that the reconstruction loss played a diminishing role, which is a very likely explanation of the observed mode collapse behavior. This becomes obvious when the loss described above is written in a more standard form

$$\text{Loss} = \frac{1}{D}\text{KL}(q(z,|x)||p(z)) - \frac{1}{L}\log(p(x,|z)) = \quad (3)$$

$$= \frac{1}{L}\left(\frac{L}{D}\text{KL}(q(z,|x)||p(z)) - \log(p(x,|z))\right) = \quad (4)$$

$$= \frac{1}{L}\left(\frac{L}{D}\sum_{i=1}^{D} -0.5\left(1 + \sigma_{\log,i} - \mu_i^2 - e^{\sigma_{\log,i}}\right) - \sum_{i=1}^{L} y_i \log(\hat{y}_i) + (1 - y_i)\log(1 - \hat{y}_i)\right) \quad (5)$$

Hence, the KL part is multiplied by $L/D = 244$ before the normalization by $L$ (that could then also be dropped). When introducing the 0.01 scaling factor, we implicitly set $L/D$ close to 1 (0.01*3906/16 = 2.44, instead of 244 as before):

$$\text{Loss} = \frac{1}{L}\left(\frac{0.01L}{D}KL(q(z,|x)||p(z)) - \log(p(x,|z))\right) \quad (6)$$

This most likely enabled to reconstruction loss to play a significant role in the optimization process and solved the mode collapse problem. We thank the anonymous reviewer for catching our overcomplication and hope this explanation will help researchers using similar models.

## VAE evaluation

The sequence logo seen in Fig. 2b was generated using the Python library logomaker[63]. Amino acid frequency and pairwise amino acid frequency (Supplementary Fig. 2a) were calculated with a custom python script, gaps were not taken into account. The projection on the latent space (Supplementary Fig. 2b) was generated by encoding all 5,958 natural sequences in the latent space, calculating a Principal Component Analysis of the resulting 16-dimensional vector representations, and projecting on the first two principal Components. In the scatterplot in Supplementary Fig. 2b only the four largest phylogenetic groups are displayed for clarity.

## Sequence generation and selection

With the final VAE model, 4,000 sequences were generated by randomly sampling from a 16-dimensional normal distribution and passing these draws through the Decoder. An amino acid sequence was then generated by taking the maximum value of the 21 possible letters (20 amino acids plus gap) at each position of the MSA. As we wanted to make homologs distant to MinE in *E. coli* (ecMinE), we excluded all generated sequences that had >=60% identity with ecMinE (UniProt ID P0A734). Then, to ensure heterogeneity among the sequences to test, we clustered the remaining sequences by 60% identity using cd-hit[58] and randomly selected one sequence per cluster for further analysis. 167 sequences remained. All pairwise sequence alignments (Fig. 4e, Supplementary Data 1) were calculated using browser version of the EMBOSS Needle pairwise sequence alignment tool[64] with default parameters.

## in silico function estimation

The emergent function of MinE is known to be based on three properties[22–24]: (i) membrane binding, which is mediated by a short hydrophobic N-terminal alpha helix, (ii) stimulation of MinD's ATPase activity by formation of a MinD-MinE heterodimer, for which a conformational switch in MinE, changing a beta-sheet to an alpha-helix, is needed, and (iii) the formation of homo-dimers. As we eventually wanted to score the generated sequences for the emergent function, we defined individual scores to estimate the capability to show each of the three individual properties, and then summed them up to a final Function Score. As preparation, we used AlphaFold Multimer[30] to predict the structures of the generated MinE homologs under two conditions: first, in presence of *E. coli*'s MinD (UniProt ID P0AEZ3), thus testing for heterodimer capability, second, in presence of itself, thus testing for homodimer capability. MSAs were generated for every sequence by the AlphaFold pipeline, specifying --db-preset=full_dbs and --max_template_date = "2021-11-01". Databases were downloaded on April 19, 2022.

## Membrane binding estimation

To evaluate the membrane binding capability, we calculated hydrophobicity with ProteinSol Patches[31], using the predicted heterodimer structure as input, as MinE must be able to bind the membrane while binding to MinD. We then calculated a single score by averaging the hydrophobicity score over all amino acids of the N-terminal alpha helix. If the N-terminal region was unstructured, we averaged over the full N-terminal region, stopping at the MinD interaction helix. Supplementary Fig. 3c shows a histogram of the resulting scores.

## MinD interaction estimation

We evaluated the potential to interact with MinD based on the Predicted Align Error (PAE) matrix provided by AlphaFold Multimer. The PAE is a measure of confidence of AlphaFold Multimer[30], where small values indicate high confidence. Thus, if AlphaFold Multimer is confident about the interaction of two proteins, this value is low on average[4,30]. The interaction of MinE and MinD is known to be located at a specific alpha-helix of MinE[24], and only this part of MinE is in a stable position relative to MinD, whilst the rest of MinE has some flexibility. Thus, we evaluated the potential to interact with MinD by averaging the PAE between the MinD-binding alpha helix of each generated MinE homolog and structured regions of MinD, while neglecting other parts of the MinE structure. Supplementary Fig. 3c shows a histogram of the resulting scores.

## Dimerization estimation

We evaluated the potential to form homodimers similar to the MinD interaction, based on the PAE. We calculated the average PAE between structured regions of two MinEs, that is, alpha-helices and beta-sheets. As can be seen in Supplementary Fig. 3c, the PAE for dimerization was on average lower than for MinD interaction, indicating that most generated MinE homologs might dimerize, but not all of the might interact with MinD, or might interact correctly.

## Solubility prediction

As we eventually wanted to test the homologs in vivo in *E. coli*, we used ProteinSol[35] to predict the solubility in *E. coli*. Values above 0.7 indicate a high probability of being soluble. As can be seen in Supplementary Fig. 3c, most homologs were predicted to be soluble.

## Final Scoring and selection for in vitro screening

To merge the four individual scores (membrane binding, MinD interaction, Dimerization, Solubility) to one Function Score, we normalized each score by setting the lowest value to 0 and the highest value to 1 (for MinD interaction and Dimerization we then calculated 1-score, because good values in their metrics are small), and then summed them up, such that the final function score could reach values between 0 and 4. We then selected the highest scoring 24 and the lowest scoring 24 sequences for in vitro analysis (Fig. 2a), double-blinded them, and named them synMinEv1–48 (Supplementary Data 1, also find a table combining all computational results in the Source Data.).

## Preparation of synMinE gene library

The amino acid sequences of each synMinE variant were reverse-translated into DNA sequences using the Codon optimization tool from Integrated DNA Technologies (Coralville, IA USA) to optimize codon usage by referencing *Escherichia coli* K12. Then, the sequence of the first 30 bp (10 amino acids) were further altered by maximizing the frequency of A and T bases while keeping the translated amino acids to optimize the cell-free expression yield. Then, 5' and 3' additional sequences (Supplementary Table 1) coding T7 promoter, Ribosome binding site, T7 terminator etc. were further attached to the synMinE sequences. The resultant 48 sequences were synthesized using eBlocks Gene Fragments service (Integrated DNA Technologies).

## Estimation of cell-free expression yield of synMinE variants

Cell-free expression of synMinE variants was carried out using PUREfrex 2.0 (PF201-0.25-5-EX, GeneFrontier, Chiba, Japan) according to the instruction from the supplier. Each synMinE gene was mixed in PURE solution at 1 ng/μL, together with 4.4% of FluoroTect GreenLys in vitro Translation Labeling System (L5001, Promega, Madison, WI, USA) and then incubated for 4 h at 37 °C. Synthesized synMinE variants were then separated by sodium dodecyl sulfate poly-acrylamide gel electrophoresis (SDS-PAGE) and fluorescence was detected using Amersham Imager 600 (GE HealthCare, Chicago, IL, USA). The relative expression yield of each variant was analyzed using the Fiji software[65].

## In vitro screening assay for functional synMinE variants

synMinE variants were synthesized using PUREfrex 2.0 as described in the previous section but incubating only 1 h at 37 °C without FluoroTect GreenLys in vitro Translation Labeling System. Then, expressed synMinE solutions were 5-folds diluted in the Reaction buffer (50 mM Tris-HCl, pH 7.5, 150 mM GluK, 5 mM GluMg) and further mixed with 1 μM EGFP-MinD, 2.5 mM ATP, and 10 g/L BSA to obtain the inner solution for the assay. The concentration of synMinE solutions was further varied up to 1- to 20-folds after checking the dynamics of MinD inside the lipid droplets at 5-folds dilution to confirm the emergence of Min waves at different concentration ranges of synMinE. As a positive control, purified wtMinE (see Purification of synMinE variants) (0.75 μM) was mixed with MinD, ATP, and BSA as mentioned above.

To prepare the lipid-oil mixture, 1-palmitoyl-2-oleoyl-glycero-3-phosphocholine (POPC) (850457, Avanti Polar Lipids, Alabaster, AL, USA) and 1-palmitoyl-2-oleoyl-sn-glycero-3-phospho-(1'-rac-glycerol) (POPG) (840457, Avanti Polar Lipids) were mixed at 70:30 (POPC:POPG) mol% in chloroform at 25 g/L. Then, 50 μL of the POPC:POPG mixture was dried under nitrogen gas stream, and subsequently, 10 μL of decane (D0011, TCI Deutschland GmbH, Eschborn, Germany) and 500 μL of mineral oil (HP50.1, Carl Roth GmbH, Karlsruhe, Germany) were added to the lipid film, and lipids were resuspended in oil by vortexing for 1 min at room temperature. Then, 1 uL of the inner solution was added to 50 uL of the lipid-oil mixture and subsequently, emulsified by tapping to obtain lipid droplets.

For the observation of Min protein dynamics, 1 uL of the droplet solution was added in a well of a 384-well plate together with 50 uL of the lipid-oil mixture. Imaging of samples was carried out by a Zeiss LSM780 confocal laser scanning microscope using a Plan-Apochromat 20x/0.80 air objective (Carl Zeiss AG, Oberkochen, Germany), using a 488 nm Argon laser for excitation with 10 s intervals for 3–5 min to validate the self-organization dynamics of Min waves as previously reported[39]. The visualization of images including kymographs was carried out using Fiji software and ImageJ macro published in ref. 40. The in vitro screening scores were manually calculated as the occurrence of Min waves (the number of droplets that exhibit Min oscillations vs. the number of all droplets within the field) with every 14 positive variants, one negative variant (v4), and wtMinE as a positive control.

## Construction of plasmids encoding synMinE variants for in vivo observation and purification

All PCR fragments were amplified by using Phusion High-Fidelity DNA Polymerase (F530L, Thermo Fisher Scientific, Waltham, MA, USA) with a set of origo primers (Supplementary Table 1) and subsequently treated with DpnI (FD1703, Thermo Fisher Scientific). All 14 positive variants (v5, v10, v25, v26, v29, v31, v33, v35, v37, v40, v43, v44, v46, v48) found by the in vitro assay and 5 variants which scored the highest in silico scores but were negative in in vitro assay (v2, v9, v19, v30, v45) were cloned in a pMLB plasmid together with mGreenLantern-MinD as previously reported[23]. Briefly, sfGFP was substituted by mGreenLantern[66] gene fragment (Integrated DNA Technologies) using GeneArt Seamless Cloning and Assembly Enzyme Mix (A14606,

Thermo Fisher Scientific) according to the supplier's protocol with PCR fragments (see Supplementary Table 1 for primers) from pMLB-sfGFP-MinD.MinE[23]. Then, MinE (wild type) was substituted with each synMinE variant using the Seamless Cloning method with PCR fragments (see Supplementary Table 1 for primers). To construct pMLB-mGreenLantern-MinD and pMLB-mGreenLantern, MinE and MinD genes were omitted from pMLB-mGreenLantern-MinD.MinE plasmid using blunt end cloning technique with a PCR fragment (see Supplementary Table 1 for primers). All enzymes for cloning (DpnI (FD1703), T4 Phosphokinase (EK0031), and T4 DNA Ligase (EL0011)) were purchased from Thermo Fisher Scientific. Additionally, all 10 positive variants with high scores of in silico screening were further cloned into a pET28 plasmid with C-terminus His-tag for purification. MinE (wild type) was substituted with each synMinE variant using Seamless Cloning with PCR fragments (see Supplementary Table 1 for primers) from pET28-MinE-His. All constructed plasmids were propagated in OneShot TOP10 competent *E. coli* (C404003, Thermo Fisher Scientific) and extracted by using NucleoBond Xtra Midi kit (Macherey-Nagel GmbH, Duren, Germany) from overnight LB culture. All gene sequences were confirmed using Sanger Sequencing Service (Microsynth AG, Balgach, Switzerland).

### In vivo phenotype characterization, analysis of cell size distribution and oscillation period

Substitution of wild type *min*E gene was carried out using *E. coli* ΔminDE strain, HL1 (Δ*min*DE *zcf*117::Tn10 *rec*A::*cat*)[23,43]. HL1 was transformed with the plasmids pMLB-mGreenLantern (as -MinDE condition), pMLB-mGreenLantern-MinD (as -MinE condition), pMLB-mGreenLantern-MinD.MinE (as wt condition), or pMLB-mGreenLantern-MinD.synMinEv5, as well as other synMinE variants (as each variant condition). Transformed HL1 cells were inoculated from glycerol stocks and incubated in LB (with 100 μg/mL ampicillin) medium overnight at 37 °C. Cells were then diluted to 1:100 in 50 mL LB (ampicillin) medium and grown at 37 °C, 180 rpm for 90–180 min. After an optical density at 600 nm (OD600) of cell cultures reached ~0.1, Isopropyl-β-D-thiogalactopyranoside (IPTG) was added to the cultures at 50 μM to induce expression of Min proteins. Cells were further incubated at 37 °C, 180 rpm for 2–4 h and then diluted in LB (ampicillin) medium with 50 μM IPTG to an OD600 of 0.1.

To prepare agarose pads, 1% (w/v) of UltraPure Low Melting Point Agarose (16520050, Life Technologies, Carlsbad, CA, USA) was first melted in LB (ampicillin) medium with 50 μM IPTG at 60 ˚C using a bench-top incubator. Then, 400 μL of agarose solution was pipetted onto a coverslip, and another coverslip was immediately placed on top of the agarose solution to obtain a planer surface of agarose pads. The agarose solution was left at room temperature for 30 min to obtain solid pads. The coverslip was removed from the top of the pad, and then cultured cells (1 μL) were spotted onto the agarose pad and left for 10 min at room temperature. Then, the agarose pad was flipped onto another coverslip and mounted to a Zeiss LSM780 confocal laser scanning microscope. Imaging was carried out as described for in vitro assay using C-Apochromat 40x/1.20 water-immersion objective (Carl Zeiss AG). The oscillation of Min proteins inside cells was captured with 5 s intervals. The observation was performed within 2 h after sample preparation at room temperature.

The phenotypes of each synMinE variant were determined as (1) the filamentous: elongated (>25 μm in length) cells were observed, (2) the minicell: miniature size (<1 μm in length) cells were observed more than 6.3% of the population (which is three times higher than wild type MinE (2.1%)), and (3) the normal. The phenotypes were confirmed to show the same morphology by at least three biological replicates. The cell size distribution and periods of Min oscillations were further analyzed by using Fiji software and custom ImageJ macro. Briefly, time-averaged fluorescence of mGreenLantern-MinD was used to determine cell position and length. Then, fluorescence

was normalized along with a long axis of the cell to obtain 1-dimensional images and then vertically stacked at each time point, resulting in a kymograph. Then the period of Min oscillation was obtained by fitting the fluorescent signals (along with the vertical (time) axis) with a sine function.

### Cell growth measurement

The *E. coli* HL1 cells transformed with pMLB plasmids were inoculated from glycerol stocks and incubated in LB (with 100 μg/mL ampicillin) medium for overnight at 37 °C. Cells were then diluted to 1:100 in 50 mL LB (ampicillin) medium and incubated at 37 °C, 180 rpm for 90 min (180 min in case -MinE and v31 conditions due to the slow cell growth) and the OD600 was measured as $t = 0$. Subsequently, 50 μM of IPTG was added to the cultures and cells were further incubated at 37 °C, 180 rpm for 5 h. The OD600 of cell cultures were continuously monitored at each 30 min after addition of IPTG, and total 11 time-points were measured per sample.

### Purification of synMinE variants

Purification of EGFP-MinD, MinD, MinE, and all synMinE variants was performed according to the previous reports[67,68]. In Brief, BL21(DE3) pLysS cells were transformed by pET28-EGFP-MinD[20,23], pET28-MinD[21], pET28-MinE[21], synMinEv5 or the other synMinE variants, and then incubated in LB medium (with 50 μg/mL Kanamycin) for overnight at 37 °C. The overnight cultures were then diluted to 1:100 in 500 mL LB (Kanamycin) and incubated while shaking at 37 °C, 180 rpm. Then, IPTG was added at 1 mM to induce overexpression of proteins once OD 600 nm reached 0.2-0.3. Cells were further cultured for 3–4 h and harvested.

The cell pellets were resuspended in Lysis buffer (50 mM Tris-HCl, pH 7.5, 300 mM NaCl, 10 mM Imidazole) and subsequently lysed using a tip sonicator (Branson ultrasonics S-250D, Thermo Fisher Scientific). The cell lysates were centrifuged for 30 min at 20,000 x g, 4 °C and then the supernatants were mixed with Ni-NTA agarose (30210, QIA-GEN, Hilden, Germany). The samples were then incubated for 10 min at 4 °C and loaded into a gravity column. Subsequently, Ni-NTA agarose was rinsed with Wash buffer (50 mM Tris-HCl, pH 7.5, 300 mM NaCl, 20 mM Imidazole, 10 % Glycerol), and then the proteins were eluted with Elution buffer (50 mM Tris-HCl, pH 7.5, 300 mM NaCl, 250 mM Imidazole, 10 % Glycerol). The buffer of the protein solution was dialyzed with Storage buffer (50 mM Tris-HCl, pH 7.5, 150 mM GluK, 5 mM GluMg, 10 % Glycerol) using Amicon Ultra-0.5 centrifugal filter unit 3 kDa (Merck KGaA, Darmstadt, Germany) and stored at −80 °C until further use. The concentration of the proteins was measured by Bradford Assay (5000006, Bio-Rad, Hercules, CA, USA), and separated by SDS-PAGE to check the purity.

### Size exclusion chromatography

The oligomerization of synMinE variants was estimated using ÄKTA pure with Superdex 75 Increase 10/300 GL column (Cytiva, Marlborough, MA, USA). The column was equilibrated with Reaction buffer prior to the measurements. The standard proteins (Blue Dextran2000: 2000 kDa, Aldolase: 158 kDa, Conalbumin: 75 kDa, Ovalbumin: 44 kDa, Carbonic Anhydrase: 29 kDa, RNaseA: 13.7 kDa, Aprotinin: 6.5 kDa) were separately loaded into the column and then eluted fractions were monitored to determine the peak fraction. The peak fraction of each standard protein was then fitted to obtain a standard curve by calculating

$$K_{av} = V_e - V_o/V_c - V_o \tag{7}$$

where $V_o$ is the column void volume (the elution volume of Blue Dextran2000), $V_e$ is the elution volume of each sample, and $V_c$ is the geometric column volume (23.5 mL in case of Superdex 75 Increase 10/ 300 GL column).

The eluted fractions of each synMinE variant were monitored and then the elution volume was determined as the peak fraction. The oligomer size of synMinE variants was then calculated from the standard curve.

## ATPase assay

ATPase assay was performed following the previous report using NADH-coupled assay[19]. To prepare small unilamellar vesicles (SUVs), 1,2-dioleoyl-sn-glycero-3-phosphocholine (DOPC) (850375, Avanti Polar Lipids) and 1,2-dioleoyl-sn-glycero-3-phospho-(1'-rac-glycerol) (DOPG) (840475, Avanti Polar Lipids) were mixed at 70:30 mol% in chloroform at 25 mg/mL. Lipids were then dried under nitrogen gas stream and then hydrated in Min buffer (25 mM Tris-HCl, pH 7.5, 150 mM KCl, 5 mM MgCl$_2$) at 4 mg/mL. Subsequently, the solution was vortexed to obtain multilamellar vesicles and the solution was further extruded through a membrane with 50 nm pore size to break down to the small unilamellar vesicles.

For the measurement of MinD's ATPase activity, 0.2 mg/mL of SUVs solution, 1 mM ATP, 2 mM phosphoenolpyruvate, 0.5 mM NADH, the mixture of pyruvate kinase (600–1000 U/mL) and lactate dehydrogenase (900–1400 U/mL) (P0294, Sigma-Aldrich, St. Louis, MO, USA), 2 µM MinD, and 2 µM of MinE or each synMinE variant were mixed in the Min buffer. Then, the decrease in absorption at 340 nm was measured in a 96-well plate using the Spark multimode microplate reader (TECAN, Männedorf, Switzerland). To calculate the ATPase activity, the linear parts of the measured values of the NADH absorption were fitted to gain a linear regression curve.

## Quartz crystal microbalance with dissipation monitoring (QCMD) measurements

QCMD measurements were carried out following the previous report[23]. Prior to each measurement, silicon dioxide (SiO2)-coated quartz crystal sensors (Biolin Scientific, Gothenburg, Sweden) were treated with a 3:1 mixture of sulfuric acid and hydrogen peroxide (piranha-solution). Subsequently, sensors were rinsed with ultrapure water, dried under a stream of nitrogen, and mounted in the flow modules of the Qsense Analyzer (Biolin Scientific). After baseline stabilization, supported lipid bilayers (SLBs) formation was induced through constant injection (flow rate: 0.15 mL/min) of a 1 mg/mL mixture of SUVs solution (prepared as described in the method section for ATPase assay), in the TK buffer (20 mM Tris-HCl pH 7.5, 150 mM KCl), spiked with 5 mM CaCl2. The formed SLBs were washed with TK buffer until no frequency change was observed. Then, 150 µl of the 5 µM of each synMinE variant in TK buffer was flown over the sensor at 0.15 ml/min and the change in frequency was monitored at overtone F9. The measured frequency was normalized by averaging the value of 5 consequent measurements at each time point, and then the change in frequency was determined by subtracting the maximum value (as baseline) from the minimum value (as dropped frequency).

## Post-Hoc analysis of in silico scoring

The improved combined function score was calculated as combined_score_improved = (normalised_MinD_score + normalized_nterminal_hydrophobicity_score). To generate the ELBO-based score, we passed every synMinE sequence 200 times through the VAE, calculated the ELBO loss (see above) and averaged. To generate the HMM score, we run hmmsearch (HMMER 3.1b2 (February 2015) using the same Hidden Markov Model Profile of the MinE domain as mentioned above (used command: hmmsearch --tblout ./hmmsearch_scores.out ./minE.hmm ./synMinEs.fasta).

## Data analysis and statistics

All statistical tests were carried out using the R software, with the exception of the one sided Mann–Whitney–Wilcoxon test in the post hoc analysis, which was performed using SciPy. Welch's $t$ test (double-sided) was used for cell growth analysis as a standard unpaired $t$ test to avoid multiplicity issues. The Mann–Whitney $U$ test (double-sided) was used for cell size distribution analysis since distributions were not normally distributed, and therefore a $t$ test would not have been suitable. The sample size (shown as $n$) and biological replicates (shown as $N$) are indicated in the corresponding figures.

## Reporting summary

Further information on research design is available in the Nature Portfolio Reporting Summary linked to this article.

## Data availability

All data are available in the text, figures, or supplementary information. All source data for graphs and gel images are provided as a Source Data file. Due to their large file size (>10 GB), the original image data files for in vitro screening and in vivo screening are available from the corresponding author upon request. Source data are provided with this paper.

## Code availability

All code is available at https://github.com/BelaFrohn/synMinE.

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

## Acknowledgements
The authors would like to thank the MPIB Core Facility, Michaela Schaper, Katharina Nakel, Kerstin Röhrl, Beatrix Scheffer, and Sigrid Bauer for assistance in protein purification, plasmid cloning, in vitro characterization, preparation of competent cells, cell culturing, and lipid preparation. We are also grateful to Beatrice Ramm for her kind advice in HL1 cell preparation, to Karsten Borgwardt for useful feedback on the manuscript, and to Jürgen Cox for inspiring discussion. SK is supported by JSPS Overseas Research Fellowships. B.P.F. is supported by the Graduate School of Quantitative Biosciences Munich (QBM). LB is supported by the Max Planck School 'Matter to life'.

## Author contributions
S.K., B.P.F., and P.S. conceived the study. B.P.F. designed, performed, analyzed, and visualized all protein design and in silico screening. S.K. designed, performed, analyzed, and visualized all in vitro screening, in vivo screening, and in vitro characterization with the help from L.B. for the QCMD measurement. S.K., B.P.F., and P.S. wrote the manuscript, and all authors discussed and revised the manuscript.

## Funding

## Competing interests
The authors declare no competing interests.
