## [Peer Review File · Nature Communications]

Machine Learning-Aided Design and Screening of an Emergent Protein Function in Synthetic CellsREVIEWER COMMENTS

Reviewer #1 (Remarks to the Author):

Overall comments

The article proposes an end-to-end approach for re-design of proteins displaying complex functionalities (e.g. dependent on the cellular environment, mediated by multiple interactions) based on candidate generation with sequence generative models and multi-stage screening. While prior machine learning based protein design works have used some combination of generative models and in silico or wet lab screening, this work goes further both in choosing a more complex target function and in proposing a fairly unified end-to-end pipeline, exploiting a hierarchy of screening methodologies (in silico, in vitro and in vivo) carefully tailored to this target function. The authors' screening strategies are methodologically interesting and their results indicate that they are to some degree successful. In particular, by combining cell free expression with a minimal synthetic model of the relevant features of the cellular environment, the authors are able to generate a fairly rapid in vitro screen suitable for assaying minE's cellular environment-dependent functionality, enabling identification of a novel minE variant that recapitulates native function despite diverging reasonably significantly from both the WT protein and any homologue in the MSA. Overall, the main contribution of the article, in my reading, is the demonstration that making appropriate choices at each stage of such a design pipeline (candidate generation and multi-stage screening) can lead to successful design (or rather recapitulation) of a particular complex functionality within a very reasonable experimental timeframe. From my knowledge of the ML based protein design literature in particular, I believe this result is in itself of considerable interest, and the screening methodology to be innovative and potentially influential for future design efforts targeting complex functions.

One limitation of the multi-stage candidate generation and screening approach is that it makes it challenging to clearly assess the contributions and importance of methodological choices made at individual stages. Instead all the stages lend support to each other via the consistently higher performance of proteins with high in silico scores across all screening stages. The paper would be improved if more direct, quantitative statements could be made about the effectiveness of individual screening stages, further supporting and contextualising the authors' claims to their importance. For example, the authors perform extensive additional post-hoc characterisation of the 10 high in silico scoring proteins that passed the in vitro phase. However, I missed a direct comparison between the in silico predicted scores for each sub function and the results of the in vitro sub-function specific assays. Similarly, it would perhaps help (me at least) understand the additional contribution of the in vitro phase if the authors were able to make some statement regarding how plausible it is that a candidate that did not pass this phase might have nonetheless been functional in vivo.

The article is very clearly written and motivated, and the authors' interpretations of their results are reasonable.

MSA VAE

I am best placed to assess the part of the work relating to the first stage of pipeline, in which a deep generative model trained on a set of minE homologues is used to generate novel minE candidates to be assessed during the screening process. Generating the candidates to such a screening pipeline seems a compelling use-case for evolution-based generative models, which effectively provide a way of intelligently diversifying naturally occurring sequences. The choice of candidate generation method is indeed quite well motivated in the text. It could be of interest (but by no means necessary) to comment even further on (i) the importance of respecting the evolutionary constraints captured by models such as the MSA VAE, rather than opting for a fully de-novo design strategy and (ii) the importance of achieving novelty using a generative model rather than simply substituting in a suitably chosen gene from a different organism for example. Presumably there are envisageable practical design settings where having these properties is desirable? If so it would be nice to explicitly set these out, to indicate how the approach proposed can generalise beyond 'mere' recapitulation of native function.

In their particular implementation of the MSA VAE approach, the authors make a couple of non-standard choices, which should be explained more clearly and justified as far as possible. First, despite the categorical nature of protein sequence data, the authors choose to use the binary cross entropy rather than the more standard categorical cross entropy. Second, the equation for the ELBO loss provided by the authors is non-standard (compare to e.g. Hoffman equation 7 or Kingma equation 10, paying attention to the meaning of L !). The authors choose to normalise the reconstruction term by the number of binary variables ($L = C \times K$, where C is the number of MSA columns and K is the number of amino acids); whereas they normalise the KL term by the number of latent dimensions, D . This inconsistent normalisation is quite possibly the source of the reported 'mode collapse' issues, which are in fact rarely encountered when using VAEs: in standard approaches, either both terms are unnormalised, or equivalently they are normalised by dividing by a single common constant. The effect of the authors' choice of normalisation is to make the KL term much larger (by a factor of L/D) than it would be in a standard VAE. In detail, supposing we choose to use the standard (negative) ELBO (Kingma eq 7):

$$J = \text{KL}(q(z|x) || p(z)) - \log p(x|z)$$

We then divide both terms by L :

$$J' = \text{KL}/L - \log p(x|z)/L$$

The authors' expression is instead:

$$J' = \alpha \text{KL}/L - \log p(x|z)/L$$

Where α is $0.01L/D = 0.01 \times 3906 / 16 = 2.44$

Thus the authors effectively upweight the KL term, which might have the effect of causing underfitting. Indeed without applying the additional rescaling by the factor 0.01, the authors are unweighting the KL term by a factor of 244 which might prevent the model from storing much information in the latent space and explain the reported mode collapse.

At the very least, the MSA VAE methods section should be re-written so that the loss is stated in a standard form (e.g. roughly as above but more explicit!), and the explanation of the necessity for the 0.01 scaling factor should be reconsidered or removed. It could also be interesting to see whether using a standard alpha of 1 improves the authors' chosen metric: i.e. correlation of pairwise frequencies between the training and generated sequences, which is currently suspiciously low (for comparison in the original MSA VAE paper the correlation of pairwise frequencies was 0.96, although there are other differences in the data and the way it is processed).

Since the use of the MSA VAE is not a particular methodological contribution of the article, I do not believe that any deficiencies in the implementation here invalidate the overall contributions. However, non-standard design choices should at minimum be clearly stated, if not justified more strongly.

Other comments

It would be interesting to provide further post-hoc analysis of association between in silico metrics and experimental success. Do the in silico scores considered, or other scores like sequence identity to WT or nearest MSA member, or likelihood of the sequence under the MSA VAE, successfully prioritise functional sequences, sequences which induce the normal cellular phenotype, or sequences with high wave occurrence percentage in the in vitro screen. Perhaps most importantly, does v25 stand out in any of these, or other, metrics?

The pairwise amino acid frequency is perhaps not the best choice for assessing the extent of MSA VAE fit. In particular, the pairwise covariance described in the MSA VAE paper more clearly distinguishes the performance of complex generative models from naive baseline strategies based on profile models, which by construction are incapable of generating data with non-zero covariances (not true for pairwise frequencies).

The authors on multiple occasions distinguish between respecting evolutionary constraints and 'introducing random noise'. This is a slightly non-specific phrasing that I find confusing and the authors may wish to re-consider. The point is to accurately capture the distribution of the sequences, and models will do this more or less well, without every risking producing true 'noise'.

For clarity, it would be helpful if the authors explained in further detail how sequences were sampled from the MSA VAE. Presumably they passed random draws from the prior through the decoder? But did they take the argmax of the softmax probabilities or sample from them to determine amino acids at each site? Furthermore, presumably gaps were excised before any further analysis but I don't think this was stated explicitly.

The motivation for choosing to in vitro screen the worst scoring as well as best scoring sequences after in silico screening is not very clear (e.g. even if AF misprediction was responsible for lower scores, why should the lowest overall scoring proteins be expected to perform better than proteins with scores slightly lower than the best scoring set.) But this choice is beneficial for the lens it provides on the effectiveness of the in silico stage.

What MSAs were used for AlphaFold Multimer?

As a practical point, in case others are interested in using the data from this study, it would be useful to provide all scores from each stage of the pipeline (where applicable) together with the corresponding sequences in a single CSV file, and to provide this in the supplementary data as well as the repo. From a brief look at the Data files in the repo it wasn't easy to tell whether such a file exists. I'd recommend documenting the data files provided in a readme file.

Reviewer Expertise

I am not well placed to assess the novelty of the in vitro screening protocol or to review the technical details of the wet lab experiments and associated results. I have correspondingly focussed my review mostly on aspects of in silico methodology and analysis, and overall contributions.

References

Diederik P. Kingma and Max Welling (2014), "Auto-Encoding Variational Bayes", International Conference on Learning Representations, <https://openreview.net/forum?id=33X9fd2-9FyZd>

Matthew D. Hoffman and Matthew J Johnson (2016), "ELBO surgery: yet another way to carve up the variational evidence lower bound", Advances in Approximate Bayesian Inference, <http://approximateinference.org/accepted/HoffmanJohnson2016.pdf>

Reviewer #2 (Remarks to the Author):

The paper explores the validation of artificially derived amino acid sequences based on machine learning, specifically focusing on their functionality in both in vitro and in vivo settings. The study employs the Multiple Sequence Alignment based Variational Autoencoder method to select around 4000 protein candidates from MinE protein. Among these, the authors further selected proteins with low identity to the wild-type and narrows down to 24 high-scoring and 24 low-scoring candidates based on four features: membrane binding, homodimer formation, heterodimer formation, and solubility. The validation process involves several in silico analyses such as AlphaFold 2 multimer, which were followed by in vitro and in vivo validations.

Out of the 48 proteins examined, 14 exhibited functional behavior in in vitro assays. Furthermore, while distinct from wild-type, several proteins demonstrated MinE functionality in in vivo experiments. Notably, the protein variant 25 displayed nearly equivalent functionality to wild-type MinE in both in vitro and in vivo contexts. This highlights the potential of synthetic amino acid sequences, derived computationally, to generate functional proteins.

While the approach utilized for protein derivation may not be entirely novel, this study's distinctive focus on experimental validation offers a valuable contribution not only to the field but also to a broader readership. Addressing concerns related to the bottleneck associated with synthetic sequence derivation using computer design, this study underscores the importance of experimental verification. Although I have some comments on this study as described below, I basically agree with the publication in Nature Communications.

Comments:

1. There is a description that 'All four scores were normalized and summed up, resulting in a roughly normally distributed final "Function Score" (Fig. 2a lower panel).' (138-140). Related to this, I'm interested in which in silico assessments actually influenced on the final MinE functions and I'd like to see the discussion on this matter.

2. Expression levels of MinE were not considered in the in silico assessments but expression levels differed among the selected variants. I'm interested in How does the stoichiometry with MinD affect the final MinE functions and I'd like to see the discussion.

Reviewer #3 (Remarks to the Author):

The goal of this research is to design and validate novel ML (machine learning)-designed artificial homologs of a protein that can interact with other proteins to produce spatial temporal patterns in a bacterial cell. These ML-designed candidates then are screened by various tests (in silico-computational as well as in vitro reconstitution) and finally tested in vivo. Overall, the authors have applied a known program to suggest MinE sequences and combined it with various proven in vitro and in silico methods to test and validate the sequences. There are no real surprises here as the final sequence that complements is 75% identical to a known sequence. So the authors have established they can do this with a reasonable amount of effort and presumably down the line this will be a useful approach. Thus, this is an exercise in generating a sequence that i

The authors used a Multiple Sequence Alignment based Variational Autoencoder (from ref. 30) to generate novel MinE variants. The model is trained with ~6000 MinE sequences in the data set (MinE homologs – mostly from Gram negative bacteria). With some reassurance that evolutionary constraints are imposed, the model was used to generate 4000 variants. Sequence conservation among these 4000 matches that of the 6000 naturally occurring MinEs. The sample size of 4000 was reduced by grouping the 4000 into 167 sequence related clusters and selecting one from each. The structure of these 167 were predicted by AlphaFold 2 multimer and then each was evaluated in silico for three known essential properties of MinE – membrane binding (hydrophobicity of the N-terminus), dimerization and binding to MinD (both done using AF2). Also solubility was predicted. Candidates with good scores tended to resemble wild type MinE from *E. coli*. Of the 167, 24 with the best scores and 24 with the worst scores were selected for in vitro screening. Proteins were expressed in a cell free system and tested for ability to

induce a spatial pattern in vitro. Of the 48, 14 were capable. Of these 10 came from the high scoring group and 4 from the low scoring group. These were then tested for in vivo oscillations and complementation. Eight of these produced oscillations in vivo even though they did not correct morphology, although cell length was reduced for seven of these compared to cell containing only MinC and MinD (filamentous). Seven resulted in filamentous morphology and grew poorly indicating inability to counteract MinCD. One (v25) was able to complement a MinE mutant to produce a WT phenotype whereas the others were less able to function.

Comments:

I did a blast with v25 and found that it was 75% identical to MinE from the gammaproteobacteria *Lysobacter*. Most of the differences were at the N-terminus (8 of the first 11 residues contained mismatches; 14 other mismatches were spread throughout the sequence but appear to occur mostly in the loops in the structure. So overall the mismatches are mostly in regions of MinE that are nonconserved or in loops as one would naively expect. Thus, I am not sure what one has learned from this exercise.

Also, there is no discussion about the residues that are known to be critical: A18, R21, I24 etc. They appear to be conserved.

Line 146-9. No references for this.

No emphasis on what is present in those 8 sequences that induce oscillation.

MinE from *Neisseria* is reported to induce oscillation along with *E. coli* MinD (<https://doi.org/10.1046/j.1365-2958.2002.03168.x>) and it only has 50% identity with *E. coli* MinE.

Reviewer #4 (Remarks to the Author):

This manuscript by Schwille and colleagues describes a new protein design platform, with validation in experimental data.

The authors highlight the remarkable progress in machine learning that has paved the way for designing proteins with novel biological functions. This work presents a comprehensive pipeline that incorporates in silico, in vitro, and in vivo screening methods. The researchers successfully generated 15 synthetic variants of a bacterial protein, which exhibited distinctive spatiotemporal patterns. In addition, the paper demonstrates the successful replacement of the wildtype gene with one of the machine learning-designed homologs in *Escherichia coli*, underscoring the potential of machine learning for engineering cellular functions.

Although the manuscript highlights the exciting potential of machine learning in protein design, it fails to acknowledge the possible limitations associated with its use. While machine learning algorithms can generate highly accurate predictions, their lack of explainability makes it challenging to comprehend the reasoning behind specific predictions. Consequently, this lack of transparency hinders the broader application of machine learning as a universal tool for rational protein variant design across diverse cellular contexts.

It is also worth noting that machine learning algorithms often require a substantial amount of data to generate accurate predictions. In the context of the screening pipeline described in the manuscript, the generalizability of the method remains uncertain due to limited data availability for understanding the structure-function relationships in many essential protein families. The risk of overfitting, where a machine learning model becomes overly complex and fails to effectively generalize new data, is noteworthy. This issue may arise when applying the system to novel protein classes, especially those with catalytic activity.

Another important aspect that the authors fail to address is algorithm bias and potential measures to mitigate it. Machine learning algorithms can inadvertently adopt biases present in the training data, underscoring the necessity of careful consideration and steps to address any inherent biases that may emerge during protein design using these methods.

While the authors present a compelling application of their platform using the cytoskeleton protein MinE, it remains a highly specialized example. In order to assert the platform's generality, it would be essential to demonstrate its efficacy on at least one catalytic protein, which performs actions on trans substrates such as small molecules or other macromolecules. Ideally, showcasing the platform's capabilities across multiple unrelated functional domains would further strengthen its credibility. For instance, an additional demonstration showcasing de novo design of a protein that binds nucleic acids or lipids, in conjunction with the previously mentioned example, would enhance the persuasiveness and versatility of the platform.

An important aspect of the platform's performance concerns its applicability to multi-domain proteins. Since many biologically relevant proteins consist of multiple domains, it is crucial to address how the platform performs in this context. This consideration should be acknowledged and discussed within the manuscript, as it is relevant to the broader understanding and potential application of the platform to real-world scenarios.

The rationale behind using PURE system for testing the new mutants, considering the ultimate goal of implementing them in *E. coli*, raises some intriguing questions. Whole cell lysate, as an alternative in vitro system, may offer certain advantages over PURE system. Unlike PURE system, whole cell lysate is known to provide a more comprehensive environment for protein folding and expression efficiency. It could potentially yield more informative and robust results, making it an attractive option for optimizing the in vitro pipeline. Worth noting is the cost factor, as the widespread use of this platform would involve substantial testing expenses, particularly when screening larger libraries. In this context, whole cell lysate becomes an even more appealing choice as it is significantly more cost-effective compared to PURE

system. Addressing these considerations within the manuscript would contribute to a comprehensive understanding and potential application of the platform in practical settings.

Utilizing the PURE system instead of whole cell lysate imposes a limitation on the size of proteins that can be tested.

In order to comprehensively evaluate the novelty of the design and the platform's performance, a critical analysis of the major structural and sequence differences between the novel MinE homolog and the wild-type protein is essential. This discussion becomes particularly important for readers who may not be familiar with the distinguishing features of Min family proteins. By highlighting these differences, readers can gain a deeper understanding of the unique characteristics and potential functional improvements offered by the designed protein variant.

Although the findings of this study show great promise, the limited demonstration provided does not yet substantiate labeling it as a universally applicable platform.

Several years ago, a related approach was reported in two separate studies (Russ, Lowery et al. 2005 Nature and Socolich, Lockless et. al 2005 Nature). These studies employed multiple sequence alignments to guide the design of non-native PDZ and WW domains.

To assess the novelty and significance of this work, a brief discussion comparing the algorithm used in this study to other high-throughput de novo protein design methods currently in use would be highly valuable. Such a comparison would provide insights into the unique aspects of this algorithm and its potential advantages or limitations in relation to existing methodologies.

Point by Point Responses

REVIEWER COMMENTS

Reviewer #1 (Remarks to the Author):

Overall comments

The article proposes an end-to-end approach for re-design of proteins displaying complex functionalities (e.g. dependent on the cellular environment, mediated by multiple interactions) based on candidate generation with sequence generative models and multi-stage screening. While prior machine learning based protein design works have used some combination of generative models and *in silico* or wet lab screening, this work goes further both in choosing a more complex target function and in proposing a fairly unified end-to-end pipeline, exploiting a hierarchy of screening methodologies (*in silico*, *in vitro* and *in vivo*) carefully tailored to this target function. The authors' screening strategies are methodologically interesting and their results indicate that they are to some degree successful. In particular, by combining cell free expression with a minimal synthetic model of the relevant features of the cellular environment, the authors are able to generate a fairly rapid *in vitro* screen suitable for assaying minE's cellular environment-dependent functionality, enabling identification of a novel minE variant that recapitulates native function despite diverging reasonably significantly from both the WT protein and any homologue in the MSA. Overall, the main contribution of the article, in my reading, is the demonstration that making appropriate choices at each stage of such a design pipeline (candidate generation and multi-stage screening) can lead to successful design (or rather recapitulation) of a particular complex functionality within a very reasonable experimental timeframe. From my knowledge of the ML based protein design literature in particular, I believe this result is in itself of considerable interest, and the screening methodology to be innovative and potentially influential for future design efforts targeting complex functions.

One limitation of the multi-stage candidate generation and screening approach is that it makes it challenging to clearly assess the contributions and importance of methodological choices made at individual stages. Instead all the stages lend support to each other via the consistently higher performance of proteins with high *in silico* scores across all screening stages. The paper would be improved if more direct, quantitative statements could be made about the effectiveness of individual screening stages, further supporting and contextualising the authors' claims to their importance. For example, the authors perform extensive additional post-hoc characterisation of the 10 high *in silico* scoring proteins that passed the *in vitro* phase. However, I missed a direct comparison between the *in silico* predicted scores for each sub function and the results of the *in vitro* sub-function specific assays. Similarly, it would perhaps help (me at least) understand the additional contribution of the *in vitro* phase if the authors were able to make some statement regarding how plausible it is that a candidate that did not pass this phase might have nonetheless been functional *in vivo*.

The article is very clearly written and motivated, and the authors' interpretations of their results are reasonable.

We thank the reviewer for a positive evaluation of our work and for the useful suggestions regarding the quantitative assessment of our screening pipeline. We agree that a more quantitative assessment improves the manuscript, and performed additional post-hoc analyses and experiments inspired by the comment. The results turned out to be very interesting, and we are grateful to the reviewer, as this improves the manuscript significantly.

To address the effectiveness of the *in silico* scoring, we performed additional post-hoc analyses: First, we tested differences in the four individual *in silico* scores (MinD binding, dimerization, N-terminal hydrophobicity, solubility) between variants that worked *in vitro* vs. variants that did not work *in vitro* for significance, using a one-sided Mann-Whitney-Wilcoxon test (Extended Data Fig. 7). This showed that the MinD binding score and the N-terminal hydrophobicity score were significantly different between variants that showed oscillations vs. variants that did not. Dimerization was not significant, but showed a trend in the expected direction. The solubility score, however, did not show the correct trend and hence seems to have worsened the overall score compared to if it was not used. We then analyzed the combined score as used for ranking in the manuscript, which was also significant (but slightly higher p-value than the individual scores for MinD binding and N-terminal

hydrophobicity, indicating that including the solubility score worsened the score slightly). As we had found that only the MinD score and the N-terminal hydrophobicity score had been significant individually, we also tested a new, improved score combining only these two features. This improved the scoring massively, and, importantly, was much better than the individual scores, indicating that indeed our divide-and-conquer approach can faithfully score the MinE emergent function. Importantly, in this score, v25, the one that worked *in vivo* eventually, is the second-highest ranked variant in total and the highest-ranked variant of the ones that worked *in vitro*. Hence, a score combining only these two features would have best estimated the emergent functionality. As the Man-Whitney-Wilcoxon test is nonparametric depending only on order, not on values, lower p-values can be interpreted as indicating better sorting for filtering. Indeed, amongst the top 5 scoring variants in the combined score(s) are 3 functional ones, whilst when scoring with individual scores only 1 or at most 2 amongst them are functional (Extended Data Fig. 7).

Second, we analyzed correlations between *in silico* scores and the post-hoc measured characteristics of selected proteins (Extended Data Fig. 14). However, we did not find large correlations here. We believe that this is due to a sampling bias: All the variants we characterized are variants that showed oscillations *in vitro*, so all of them are expected to show all three features, MinD binding, dimerization and membrane-binding. As the score we used is a simple linear one, but the phenomenon is a result of non-linear dynamics, we believe that the score we used at best is able to hint towards a binary choice, whether or not a sub-function can be unfolded. Hence, we did not expect proper correlations between our scores and properties of only-positive variants.

We added both points to the results in a newly introduced paragraph (p. 7, l. 12-17 and p. 9, l. 33-37).

To address the contribution of the *in vitro* phase, we performed additional experiments (Extended Data Fig. 10). We tested 5 variants that did not show oscillations *in vitro* but were scored highly *in silico* for *in vivo* function. All of them showed no oscillations or normal cell phenotype (and one variant could not even restore cell viability, therefore resulting in failure of obtaining transformed E.coli cells). Hence, we conclude that the *in vitro* filtering is a useful second filtering step towards *in vivo* screening, especially it considerably shortens the screening timeline (as we mentioned in the main text, we could screen all 48 sequences *in vitro* within two days). Importantly, when first sorting by the new *in silico* score as mentioned above (only combining MinD-interaction and N-terminal hydrophobicity), and then filtering *in vitro*, v25 is the top candidate. This indicates that the combination of *in silico* and *in vitro* filtering can indeed find top candidates for *in vivo* function. This point is now added in the results section (p. 8, l. 13-15).

MSA VAE

I am best placed to assess the part of the work relating to the first stage of pipeline, in which a deep generative model trained on a set of minE homologues is used to generate novel minE candidates to be assessed during the screening process. Generating the candidates to such a screening pipeline seems a compelling use-case for evolution-based generative models, which effectively provide a way of intelligently diversifying naturally occurring sequences. The choice of candidate generation method is indeed quite well motivated in the text. It could be of interest (but by no means necessary) to comment even further on (i) the importance of respecting the evolutionary constraints captured by models such as the MSA VAE, rather than opting for a fully de-novo design strategy and (ii) the importance of achieving novelty using a generative model rather than simply substituting in a suitably chosen gene from a different organism for example. Presumably there are envisageable practical design settings where having these properties is desirable? If so it would be nice to explicitly set these out, to indicate how the approach proposed can generalise beyond 'mere' recapitulation of native function.

Regarding (i): We included a sentence in the Results section explaining why we did not use a fully *de novo* design approach, and stated in the entirely changed introduction that we wanted to focus only on the screening, whilst development of a novel design method was not our goal (and would be extremely challenging). Regarding (ii): In the entirely changed introduction, we included a short section describing that what we needed to test the screening pipeline was a set of both functional and non-functional sequences. As MinE from distant organisms can still give rise to oscillations in E. coli (as Reviewer #3 also states), having used genes from distant organisms instead of novel variants

could have resulted in a set of sequences where almost all of them are functional – what would have been suboptimal to test our screening pipeline.

Regarding possible applications, we added some suggestions for future directions of the design of protein emergent functions in the discussion (p. 11, l. 9-15)

In their particular implementation of the MSA VAE approach, the authors make a couple of non-standard choices, which should be explained more clearly and justified as far as possible. First, despite the categorical nature of protein sequence data, the authors choose to use the binary cross entropy rather than the more standard categorical cross entropy. Second, the equation for the ELBO loss provided by the authors is non-standard (compare to e.g. Hoffman equation 7 or Kingma equation 10, paying attention to the meaning of L!). The authors choose to normalise the reconstruction term by the number of binary variables ($L = C \times K$, where C is the number of MSA columns and K is the number of amino acids); whereas they normalise the KL term by the number of latent dimensions, D. This inconsistent normalisation is quite possibly the source of the reported 'mode collapse' issues, which are in fact rarely encountered when using VAEs: in standard approaches, either both terms are unnormalised, or equivalently they are normalised by dividing by a single common constant. The effect of the authors' choice of normalisation is to make the KL term much larger (by a factor of L/D) than it would be in a standard VAE. In detail, supposing we choose to use the standard (negative) ELBO (Kingma eq 7):

$$J = KL(q(z|x)||p(z)) - \log p(x|z)$$

We then divide both terms by L:

$$J' = KL/L - \log p(x|z)/L$$

The authors' expression is instead:

$$J' = \alpha KL/L - \log p(x|z)/L$$

Where alpha is $0.01L/D = 0.01 \times 3906 / 16 = 2.44$

Thus the authors effectively upweight the KL term, which might have the effect of causing underfitting. Indeed without applying the additional rescaling by the factor 0.01, the authors are unweighting the KL term by a factor of 244 which might prevent the model from storing much information in the latent space and explain the reported mode collapse.

At the very least, the MSA VAE methods section should be re-written so that the loss is stated in a standard form (e.g. roughly as above but more explicit!), and the explanation of the necessity for the 0.01 scaling factor should be reconsidered or removed. It could also be interesting to see whether using a standard alpha of 1 improves the authors' chosen metric: i.e. correlation of pairwise frequencies between the training and generated sequences, which is currently suspiciously low (for comparison in the original MSA VAE paper the correlation of pairwise frequencies was 0.96, although there are other differences in the data and the way it is processed).

Since the use of the MSA VAE is not a particular methodological contribution of the article, I do not believe that any deficiencies in the implementation here invalidate the overall contributions. However, non-standard design choices should at minimum be clearly stated, if not justified more strongly.

We would like to sincerely thank the reviewer for this detailed, excellent comment. We humbly admit that our choice of MSA VAE resulted from a lack of experience in applying Machine Learning for our purposes. In hindsight, we absolutely agree that using CCE instead of BCE would have been a better choice. Regarding the normalization, apparently, with the rescaling by 0.01 we were able to get past a self-inflicted hurdle. We agree with the reviewer that we need to state these overcomplications and bad choices explicitly to help researchers in similar situations to us – biological scientists going into ML – to avoid repeating this. We added a paragraph in the methods section where we describe in

detail how we deviate from a standard implementation, just as the reviewer suggested, and describe what should be done differently in a new project using a similar model. However, as we found a “workaround” by introducing the rescaling, and especially because all the downstream wet-lab experiments are performed now, we do not see a reason to train a new model (by stating that “the use of the MSA VAE is not a particular methodological contribution of the article, I do not believe that any deficiencies in the implementation here invalidate the overall contributions.”, the reviewer also seems to have this view). In the end, we got some functional and some non-functional sequences, which is all we needed to test our screening pipeline.

Other comments

It would be interesting to provide further post-hoc analysis of association between *in silico* metrics and experimental success. Do the *in silico* scores considered, or other scores like sequence identity to WT or nearest MSA member, or likelihood of the sequence under the MSA VAE, successfully prioritise functional sequences, sequences which induce the normal cellular phenotype, or sequences with high wave occurrence percentage in the *in vitro* screen. Perhaps most importantly, does v25 stand out in any of these, or other, metrics?

As also mentioned above, in the additional analyses performed inspired by the comments of this reviewer, we found that when using the improved *in silico* score (which we only were able to find AFTER *in vitro* characterization), v25 is the 2nd best scoring variant *in silico* and the best scoring one after filtering *in vitro* (Extended Data Fig. 7). Hence, v25 stands out in this regard. Also, in the characterization using purified proteins, we found that v25 is the only variant that showed characteristics similar to the wildtype in all measured characteristics (Fig. 4), thus also in this regard v25 stands out. (It is also the 3rd most similar sequence to the *E. coli* wildtype sequence)

The pairwise amino acid frequency is perhaps not the best choice for assessing the extent of MSA VAE fit. In particular, the pairwise covariance described in the MSA VAE paper more clearly distinguishes the performance of complex generative models from naive baseline strategies based on profile models, which by construction are incapable of generating data with non-zero covariances (not true for pairwise frequencies).

In our case, we only wanted to distinguish our different models from each other, to find the best hyperparameters. Pairwise frequencies are sufficient for this, as can also be found in the original MSA VAE paper. Different from this original paper, we don't need a comparison to other baselines, as we treat it as an established method that was shown to perform better. Of course, we could also have done it with the covariance, however, we trust the original paper and simply search for the best among similar models, for which pairwise frequencies are sufficient.

The authors on multiple occasions distinguish between respecting evolutionary constraints and ‘introducing random noise’. This is a slightly non-specific phrasing that I find confusing and the authors may wish to re-consider. The point is to accurately capture the distribution of the sequences, and models will do this more or less well, without every risking producing true ‘noise’.

We rephrased these statements as they were indeed confusing.

For clarity, it would be helpful if the authors explained in further detail how sequences were sampled from the MSA VAE. Presumably they passed random draws from the prior through the decoder? But did they take the argmax of the softmax probabilities or sample from them to determine amino acids at each site? Furthermore, presumably gaps were excised before any further analysis but I don't think this was stated explicitly.

We included this information, both in results and methods. We took the argmax of every position and excluded gaps before further downstream analysis.

The motivation for choosing to *in vitro* screen the worst scoring as well as best scoring sequences after *in silico* screening is not very clear (e.g. even if AF misprediction was responsible for lower scores, why should the lowest overall scoring proteins be expected to perform better than proteins

with scores slightly lower than the best scoring set.) But this choice is beneficial for the lens it provides on the effectiveness of the *in silico* stage.

Indeed, the reason to do this is was to show the effectiveness of the *in silico* scoring. As we double-blinded the scores, we can be confident that the *in silico* screening actually was useful. We appreciate that our motivation was confusingly written and changed the results section to make this clearer.

What MSAs were used for AlphaFold Multimer?

MSAs were generated for every sequence by the AlphaFold2 script, using `--db-preset=full_dbs` and `--max_template_date="2021-11-01"`. Databases were downloaded on April 19 2022. We included this information in the Method section. At the time, the Max Planck Computing & Data Facility allowed running AlphaFold2 only via a specific script, what limited us to these options.

As a practical point, in case others are interested in using the data from this study, it would be useful to provide all scores from each stage of the pipeline (where applicable) together with the corresponding sequences in a single CSV file, and to provide this in the supplementary data as well as the repo. From a brief look at the Data files in the repo it wasn't easy to tell whether such a file exists. I'd recommend documenting the data files provided in a readme file.

We made and provided a single integrated file to summarize all the details of our results as suggested by the reviewer. We included it into the source data Excel file as a new sheet, but also as a raw tsv file in the Github repository. We thank the reviewer for a useful comment.

Reviewer Expertise

I am not well placed to assess the novelty of the *in vitro* screening protocol or to review the technical details of the wet lab experiments and associated results. I have correspondingly focussed my review mostly on aspects of *in silico* methodology and analysis, and overall contributions.

References

Diederik P. Kingma and Max Welling (2014), "Auto-Encoding Variational Bayes", International Conference on Learning Representations, <https://openreview.net/forum?id=33X9fd2-9FyZd>
Matthew D. Hoffman and Matthew J Johnson (2016), "ELBO surgery: yet another way to carve up the variational evidence lower bound", Advances in Approximate Bayesian Inference, <http://approximateinference.org/accepted/HoffmanJohnson2016.pdf>

Reviewer #2 (Remarks to the Author):

The paper explores the validation of artificially derived amino acid sequences based on machine learning, specifically focusing on their functionality in both *in vitro* and *in vivo* settings. The study employs the Multiple Sequence Alignment based Variational Autoencoder method to select around 4000 protein candidates from MinE protein. Among these, the authors further selected proteins with low identity to the wild-type and narrows down to 24 high-scoring and 24 low-scoring candidates based on four features: membrane binding, homodimer formation, heterodimer formation, and solubility. The validation process involves several *in silico* analyses such as AlphaFold 2 multimer, which were followed by *in vitro* and *in vivo* validations.

Out of the 48 proteins examined, 14 exhibited functional behavior in *in vitro* assays. Furthermore, while distinct from wild-type, several proteins demonstrated MinE functionality in *in vivo* experiments. Notably, the protein variant 25 displayed nearly equivalent functionality to wild-type MinE in both *in vitro* and *in vivo* contexts. This highlights the potential of synthetic amino acid sequences, derived computationally, to generate functional proteins.

While the approach utilized for protein derivation may not be entirely novel, this study's distinctive focus on experimental validation offers a valuable contribution not only to the field but also to a broader readership. Addressing concerns related to the bottleneck associated with synthetic sequence derivation using computer design, this study underscores the importance of experimental

verification. Although I have some comments on this study as described below, I basically agree with the publication in Nature Communications.

We appreciate the reviewer's agreement on the importance of our proof-of-principle pipeline to the broad audience. We also believe that the thoroughly revised introduction and discussion will do even better in this respect.

Comments:

1. There is a description that 'All four scores were normalized and summed up, resulting in a roughly normally distributed final "Function Score" (Fig. 2a lower panel).' (138-140). Related to this, I'm interested in which *in silico* assessments actually influenced on the final MinE functions and I'd like to see the discussion on this matter.

We gave a detailed answer to this question in the general remarks and response to reviewer #1 with a new post-hoc analysis to estimate the effectiveness of our screening pipeline. We believe this to be a very important point to further improve the strength of our manuscript and therefore would like to thank the reviewer for the suggestion.

2. Expression levels of MinE were not considered in the *in silico* assessments but expression levels differed among the selected variants. I'm interested in How does the stoichiometry with MinD affect the final MinE functions and I'd like to see the discussion.

To the best of our knowledge, it is still a difficult task to predict the expression level of *de novo* proteins computationally. However, in the *in silico* screening, we estimated the solubility score of each synMinE variant, which indirectly suggests the "expression level" of proteins as we eventually need soluble but not aggregative proteins as "functional" proteins. Especially, since the ProteinSol tool used for solubility prediction was developed (Hebditch et al., 2017, Bioinformatics) based on experimental analysis of protein solubility scores by using PURE cell-free expression (Niwa et al., 2009, PNAS), we initially thought that this *in silico* score might hint to expression levels. However, as described above, we were surprised to see in the post-hoc analysis that expression level and solubility score did only marginally correlate. In addition, since synMinE variants are relatively small proteins (around 9 kDa) quite high expression quantities have been expected (in the best scenario, one could expect more than 40-50 μM of synMinE, considering that PURE can yield 15-20 μM of GFP (27 kDa)), a more than sufficient concentration to invoke Min wave dynamics (discussed in the next paragraph).

Regarding the stoichiometry of MinDE protein functions, it has been known that there are two essential features necessary to yield MinDE waves within lipid droplets (Kohyama et al., 2020, Nanoscale): 1) both MinD and MinE concentration must be higher than the critical concentration (typically higher than 0.5 μM), and 2) the molar ratio of MinD and MinE must be balanced (typically MinD:MinE = 1:1 ratio). Importantly, regarding the second point, it was also shown that the effective MinE concentration was drastically different when MinE mutants were used for the assay, where 5-10 fold lower/higher concentration range is required to yield Min waves (Kohyama et al., 2020, Nanoscale). Therefore, in this study, we fixed the MinD concentration at 1 μM as the standard condition and then differed the dilution level of expressed synMinE solution up to -20 fold to validate a wide concentration range (as has been stated in the method section). Together with the fact that we had more than 80% success of detectable cell-free expression among all variants and even two non-detectable variants induced Min waves *in vitro*, we concluded that our *in vitro* screening covered a wide stoichiometric range for MinE protein function.

Reviewer #3 (Remarks to the Author):

The goal of this research is to design and validate novel ML (machine learning)-designed artificial homologs of a protein that can interact with other proteins to produce spatial temporal patterns in a bacterial cell. These ML-designed candidates then are screened by various tests (*in silico*-computational as well as *in vitro* reconstitution) and finally tested *in vivo*. Overall, the authors have applied a known program to suggest MinE sequences and combined it with various proven *in vitro*

and *in silico* methods to test and validate the sequences. There are no real surprises here as the final sequence that complements is 75% identical to a known sequence. So the authors have established they can do this with a reasonable amount of effort and presumably down the line this will be a useful approach. Thus, this is an exercise in generating a sequence that i

The authors used a Multiple Sequence Alignment based Variational Autoencoder (from ref. 30) to generate novel MinE variants. The model is trained with ~6000 MinE sequences in the data set (MinE homologs – mostly from Gram negative bacteria). With some reassurance that evolutionary constraints are imposed, the model was used to generate 4000 variants. Sequence conservation among these 4000 matches that of the 6000 naturally occurring MinEs. The sample size of 4000 was reduced by grouping the 4000 into 167 sequence related clusters and selecting one from each. The structure of these 167 were predicted by AlphaFold 2 multimer and then each was evaluated *in silico* for three known essential properties of MinE – membrane binding (hydrophobicity of the N-terminus), dimerization and binding to MinD (both done using AF2). Also solubility was predicted. Candidates with good scores tended to resemble wild type MinE from *E. coli*. Of the 167, 24 with the best scores and 24 with the worst scores were selected for *in vitro* screening. Proteins were expressed in a cell free system and tested for ability to induce a spatial pattern *in vitro*. Of the 48, 14 were capable. Of these 10 came from the high scoring group and 4 from the low scoring group. These were then tested for *in vivo* oscillations and complementation. Eight of these produced oscillations *in vivo* even though they did not correct morphology, although cell length was reduced for seven of these compared to cell containing only MinC and MinD (filamentous). Seven resulted in filamentous morphology and grew poorly indicating inability to counteract MinCD. One (v25) was able to complement a MinE mutant to produce a WT phenotype whereas the others were less able to function.

We appreciate the constructive comments by the reviewer. As mentioned, it is not surprising that synMinE sequences have high similarity to the existing natural homologs since we have utilized the MSA-VAE to generate “synthetic homologs” that are intended to be “similar but different” sequences. As we already discussed in the general remarks, here we present a proof-of-concept for how to screen for emergent functions, but do not aim to fully *de novo* design a protein with emergent function. This will be the subject of subsequent studies inspired by this manuscript. Here, we show in the *in silico* part that a scoring method which, importantly, does not take into account sequence similarity can successfully screen for emergent function (see also the new post-hoc analysis, especially Extended Data Fig. 8, where it is shown that the function score does only marginally correlate with sequence similarity to natural proteins). We believe that this will be of great use to subsequent fully *de novo* design projects, where scoring based on sequence similarity is not possible. Regarding the rest of point by point responses and the revised manuscript, we hope the reviewer agrees with us.

Comments:

I did a blast with v25 and found that it was 75% identical to MinE from the gammaproteobacteria *Lysobacter*. Most of the differences were at the N-terminus (8 of the first 11 residues contained mismatches; 14 other mismatches were spread throughout the sequence but appear to occur mostly in the loops in the structure. So overall the mismatches are mostly in regions of MinE that are nonconserved or in loops as one would naively expect. Thus, I am not sure what one has learned from this exercise.

Also, there is no discussion about the residues that are known to be critical: A18, R21, I24 etc. They appear to be conserved.

We thank the reviewer for requesting a better discussion of the final outcomes, which allowed us to provide better insight into our study. We added an analysis of the MSA of the synMinE variants tested *in vitro*, and found that MinE’s crucial residues for MinD ATPase stimulation as well as for its conformational changes are conserved in almost all variants (Extended Data Fig. 15). This should not be a surprise, because this is exactly what we find in natural homolog MSAs and even in *in vitro* directed evolution/engineering of proteins, namely, that the core residues for protein function cannot or only very limited be replaced with other amino acids (therefore they are evolutionally conserved),

whilst the “scaffold” part can be altered to optimize the protein for its cellular environment (which is highly dependent on the species). Hence, conserving the core residues while varying the scaffold residues in a way that the core ones are still correctly arranged in the three-dimensional structure is exactly what the MSA-VAE is supposed to learn, and that we see this conservation in the generated sequences is a sign that the sequence generation indeed worked as intended. In addition, although crucial residues are conserved among most of the synMinE variants, we ended up with only one variant that fully restored its *in vivo* function. Furthermore, there are generated sequences that conserve all the core residues and are predicted to fold very similar to the natural MinE, but nonetheless do not even give rise to oscillations *in vitro* – and this is correctly predicted by the *in silico* scoring. This provides a hint that indeed the *in silico* scoring did capture some degree of function and not only sequence similarity. These results are now included in the Results section (Extended Data Fig. 15, p. 10, l. 1-9).

Line 146-9. No references for this.

We apologize for this shortcoming. We included an appropriate reference (Ramm et al., 2019, Cellular and Molecular Life Sciences) here.

No emphasis on what is present in those 8 sequences that induce oscillation.

Please see the second point by point response above. Indeed, all essential residues are conserved among all 8 positive sequences which is not surprising, however, it is difficult to distinguish the critical features between these 8 sequences and other non-functional sequences, where also all critical residues are conserved. Those points are summarized in the results section (Extended Data Fig. 15, p. 10, l. 1-9).

MinE from *Neisseria* is reported to induce oscillation along with *E. coli* MinD (<https://doi.org/10.1046/j.1365-2958.2002.03168.x>) and it only has 50% identity with *E. coli* MinE.

This is also not surprising at all, to find a 50% identical homolog (similar to synMinEv25) among natural MinEs, since it has evolved over the course of natural selection as a functional and “wave inducible” protein. Importantly, in the paper kindly provided by the reviewer there are two major differences distinguishing the *Neisseria* MinE (MinEng) from our synMinEv25: First, the wildtype MinEng could not induce waves with GFP-MinD as we used in this study, but only the combination of MinDec and MinEng-GFP, in which GFP conjugation to the MinE may cause unexpected interferences/artifacts (therefore we did not adapt this in our study). Second, the appearance of dynamic waves with MinEng was significantly lower (70% among observed cells, in contrast to almost 100% with synMinEv25), and oscillation period is about 4 times slower (174 sec) compared to the MinEec, while synMinEv25 shows only less than 10% of difference (39 sec vs 42 sec). Therefore, we believe that even this natural homolog could not fully functionally substitute the wt MinEec in the same way synMinEv25 can. However, this is indeed another critical point to further emphasize our achievement with synMinE that we ignored before, and therefore this remark is now included in the manuscript (p. 12, l. 20-27). We are grateful to the reviewer for an insightful comment. In addition, most importantly and as mentioned above, the focus of this study is not to make any lowest identity homologs of MinE, but rather to develop a novel screening pipeline for emergent protein functions. We hope now this point is more obvious in the revised manuscript.

Reviewer #4 (Remarks to the Author):

This manuscript by Schwille and colleagues describes a new protein design platform, with validation in experimental data.

The authors highlight the remarkable progress in machine learning that has paved the way for designing proteins with novel biological functions. This work presents a comprehensive pipeline that incorporates *in silico*, *in vitro*, and *in vivo* screening methods. The researchers successfully generated 15 synthetic variants of a bacterial protein, which exhibited distinctive spatiotemporal patterns. In addition, the paper demonstrates the successful replacement of the wildtype gene with

one of the machine learning-designed homologs in *Escherichia coli*, underscoring the potential of machine learning for engineering cellular functions.

Although the manuscript highlights the exciting potential of machine learning in protein design, it fails to acknowledge the possible limitations associated with its use. While machine learning algorithms can generate highly accurate predictions, their lack of explainability makes it challenging to comprehend the reasoning behind specific predictions. Consequently, this lack of transparency hinders the broader application of machine learning as a universal tool for rational protein variant design across diverse cellular contexts.

It is also worth noting that machine learning algorithms often require a substantial amount of data to generate accurate predictions. In the context of the screening pipeline described in the manuscript, the generalizability of the method remains uncertain due to limited data availability for understanding the structure-function relationships in many essential protein families. The risk of overfitting, where a machine learning model becomes overly complex and fails to effectively generalize new data, is noteworthy. This issue may arise when applying the system to novel protein classes, especially those with catalytic activity.

Another important aspect that the authors fail to address is algorithm bias and potential measures to mitigate it. Machine learning algorithms can inadvertently adopt biases present in the training data, underscoring the necessity of careful consideration and steps to address any inherent biases that may emerge during protein design using these methods.

We would like to thank the reviewer for raising these absolutely correct and important issues about ML, especially ML on protein data. In our case, to circumvent the sampling bias of the UniProt database, we performed clustering when selecting sequences (as described in the Methods), so we made sure that a variety of different sequences were tested and not only ones from *E. coli* which are overrepresented in UniProt (and hence produce a bias, just as the reviewer correctly remarks). Furthermore, we explicitly acknowledge limitations and potential pitfalls in the revised discussion and explicitly state problems and challenges for sub-function scoring for future projects.

While the authors present a compelling application of their platform using the cytoskeleton protein MinE, it remains a highly specialized example. In order to assert the platform's generality, it would be essential to demonstrate its efficacy on at least one catalytic protein, which performs actions on trans substrates such as small molecules or other macromolecules. Ideally, showcasing the platform's capabilities across multiple unrelated functional domains would further strengthen its credibility. For instance, an additional demonstration showcasing de novo design of a protein that binds nucleic acids or lipids, in conjunction with the previously mentioned example, would enhance the persuasiveness and versatility of the platform.

As we already discussed in the general remarks, and in accordance with the Editor, we respectfully disagree with the request to entirely redo the screening process with a different target protein, especially a catalytic protein. In this study, we aim to develop an integrated screening system for one specific emergent protein function as a proof-of-concept, but not as a development for a ready-to-use method. Indeed, such emergent functions are exceptionally complex, and therefore our pipeline would have to be adapted immensely to screen for a completely different protein/function, which is simply not feasible to do within the timeframe of even a major revision. More importantly, a pipeline inspired by our proof-of-concept is not suited to screen for catalytic function as suggested by the reviewer. There exist already a plethora of screening methods for catalytic activity (Biswas et al., 2021, *Nature Methods*; Longwell, Labanieh & Cochran, 2017, *Current Opinion in Biotechnology*, and we did not want to improve these in this manuscript. Unlike higher-order protein functions that result from the combination of multiple sub-functions such as protein complex formation, membrane binding, conformational switching, DNA binding, etc., the design of proteins with catalytic activity at the moment focuses only on a single function, the protein-substrate interaction. Hence, we do not classify this as an emergent function, and thus we believe that it is not just out of our scope to redo the screening with a catalytic protein, but that it would also create a huge misconception about our goal and our achievement to readers. However, we acknowledge that we had stimulated this misunderstanding, by not clearly enough emphasizing the goal and scope of our study in the original

manuscript, and hence, we put great effort into rewriting the introduction and discussion, as well as changing the title. We hope that the reviewer agrees with us and sees a significant improvement in the clarity of the revised manuscript.

An important aspect of the platform's performance concerns its applicability to multi-domain proteins. Since many biologically relevant proteins consist of multiple domains, it is crucial to address how the platform performs in this context. This consideration should be acknowledged and discussed within the manuscript, as it is relevant to the broader understanding and potential application of the platform to real-world scenarios.

The rationale behind using PURE system for testing the new mutants, considering the ultimate goal of implementing them in *E. coli*, raises some intriguing questions. Whole cell lysate, as an alternative *in vitro* system, may offer certain advantages over PURE system. Unlike PURE system, whole cell lysate is known to provide a more comprehensive environment for protein folding and expression efficiency. It could potentially yield more informative and robust results, making it an attractive option for optimizing the *in vitro* pipeline. Worth noting is the cost factor, as the widespread use of this platform would involve substantial testing expenses, particularly when screening larger libraries. In this context, whole cell lysate becomes an even more appealing choice as it is significantly more cost-effective compared to PURE system. Addressing these considerations within the manuscript would contribute to a comprehensive understanding and potential application of the platform in practical settings.

Utilizing the PURE system instead of whole cell lysate imposes a limitation on the size of proteins that can be tested.

Regarding the concerns in the last 3 paragraphs, we added a new part to the Discussion (p. 12, l. 3-16) to comprehensively address limitations of the PURE system, as well as possibilities to further utilize cell-free expression for protein engineering and screening by using additional supplementation or switching to lysate-based cell-free expression systems (please also see the response to reviewer #2 about related topics). Essentially, there are two reasons why we chose PURE in this study. First, MinE is a small protein (around 9 kDa) and has been expressed in PURE previously (Godino et al., 2019, Nat. Comm., Yoshida et al., 2019, Chem. Sci., Kohyama et al., 2022, Nat. Comm.), thus, we expected high yields of synMinE variants, and indeed, more than 80% of the variants were expressed at detectable level. Second, in the *in silico* screening, we estimated the solubility score of each synMinE variant by using ProteinSol (Hebditch et al., 2017, Bioinformatics), which was developed based on experimental analysis of protein solubility scores from PURE system (Niwa et al., 2009, PNAS). Hence, as a demonstration of proof-of-concept, we believe that PURE is the most consistent environment for our screening pipeline.

On the other hand, as the reviewer pointed out, further bigger, difficult-to-fold, or multi-domain proteins could be obstacles in future studies since PURE is one of the simplest cell-free expression systems, consisting of only the minimal components of the *E. coli* transcription-translation machinery and therefore often struggles with those problematic proteins. To avoid such problems, one could consider additionally supplementing protein chaperones, protein disulfide isomerase, and additional ribosomal factors to the PURE system for better folding/bonding/translation properties, where all of those components are also commercially available. Moreover, as discussed by the reviewer, lysate-based cell-free expression systems can be alternative choices for *in vitro* screening since they are known to provide a comprehensive environment for protein folding and expression efficiency due to their intrinsic factors, as well as for their significantly lower costs. However, major drawbacks of cell-lysate systems are relatively large batch-to-batch differences in protein yields, time-consuming preparation steps, and unexpected interaction between newly synthesized proteins and unknown intrinsic factors, which kept us from using them for this study. Taken together, both PURE system and cell-lysate systems have their own unique advantages, and therefore, users can adapt and further modulate the most suitable cell-free expression platform depending on their needs.

In order to comprehensively evaluate the novelty of the design and the platform's performance, a critical analysis of the major structural and sequence differences between the novel MinE homolog and the wild-type protein is essential. This discussion becomes particularly important for readers

who may not be familiar with the distinguishing features of Min family proteins. By highlighting these differences, readers can gain a deeper understanding of the unique characteristics and potential functional improvements offered by the designed protein variant.

We thank the reviewer for this comment. As we already had similar suggestions from reviewer #3, we performed an additional analysis of the MSA over the synMinE variants, which is included in the Results section (Extended Data Fig. 15, p. 10, l. 1-9, also see main Figure 4e). Please also see our responses to reviewer #3 for the insights gained from the analysis. In addition to this, we would like to highlight that what we learned from this additional exercise is indeed not about MinE's sequence/structure features, as we observed that all essential residues are simply conserved. Instead, this conservation suggests that the MSA-VAE worked as intended (also discussed in the response to the reviewer #3).

Although the findings of this study show great promise, the limited demonstration provided does not yet substantiate labeling it as a universally applicable platform.

As we discussed in the general remarks and previous responses, the development of a generalizable method is not our motivation in this study, and we took better care to explicitly emphasize this point in the revised manuscript, especially in the Introduction. We thank the reviewer for expressing that our study shows great promise and hope that our message is now clearer in the revised manuscript.

Several years ago, a related approach was reported in two separate studies (Russ, Lowery et al. 2005 Nature and Socolich, Lockless et. Al 2005 Nature). These studies employed multiple sequence alignments to guide the design of non-native PDZ and WW domains. To assess the novelty and significance of this work, a brief discussion comparing the algorithm used in this study to other high-throughput de novo protein design methods currently in use would be highly valuable. Such a comparison would provide insights into the unique aspects of this algorithm and its potential advantages or limitations in relation to existing methodologies.

In the revised Introduction, as well as in a sentence we added to the sequence generation paragraph in the Results, we motivate in more detail why we used the MSA-VAE and how this compares to other sequence generation methods. We agree with the reviewer that these two studies are in line with our manuscript in one particular aspect, which is, sequences were generated based on a MSA. The goal of our study compared to the two mentioned studies, however, is a very different one: We generated sequences to have a set of candidates to test our screening method on, whilst in the two mentioned studies the question is an evolutionary one, namely, whether the information about coevolution of amino acids contained in a MSA is sufficient to specify fold and function. Indeed, the mentioned studies showed this, which is why we could use a MSA based sequence generative model. We included the two papers as references in our motivation for the MSA-VAE.

References

- Glock et al., 2019, eLife,
- Kretschmer et al., 2017, Plos One
- Biswas et al., 2021, Nature Methods
- Longwell, Labanieh & Cochran, 2017, Current Opinion in Biotechnology
- Hebditch et al., 2017, Bioinformatics
- Niwa et al., 2009, PNAS
- Kohyama et al., 2020, Nanoscale
- Godino et al., 2019, Nat. Comm.
- Yoshida et al., 2019, Chem. Sci.
- Kohyama et al., 2022, Nat. Comm.
- Ramm et al., 2019, Cellular and Molecular Life Sciences

REVIEWER COMMENTS

Reviewer #1 (Remarks to the Author):

The authors have revised the presentation of their paper to clarify its contributions and performed additional analysis that supports the claims made for those contributions. In particular, the authors demonstrate in post-hoc analysis that the in silico and in vitro screening successfully identify the functional in vivo design as the leading candidate after the first two screening stages, providing strongly suggestive evidence that these screening methods are effective and complementary. Additional quantitative analysis further supports these claims to the extent that can be expected for this kind of proof of principle study. My previous concerns have been addressed carefully, the manuscript is improved and seems to me a valuable contribution to the field.

I have minor remaining feedback relating to the revisions. In particular some small pieces of further in silico analysis relating to in silico scoring functions could strengthen the conclusions further.

The authors could support their choice of an evolution-based generative model more strongly rather than entirely in negative terms (as a fallback due to the difficulty of creating a de novo design method). Evolution-based models such as the MSA VAE have had notable success in functional protein design and can be seen as a way of generating diversity while implicitly conditioning on function. Some of this benefit is lost in the current presentation.

The authors clearly and openly address their non-standard VAE modelling choices in response to my previous comment. This should be helpful to readers unfamiliar with the model class. I found one sentence in the revised description somewhat ambiguous: 'L is the length of the one hot encoded sequence'. It would be easy to interpret this sentence as being equivalent to saying that L is the number of residues, whereas in fact the authors mean that L is the number of binary variables after one-hot encoding the sequence and then flattening it (one-hot encoding produces a matrix whose first dimension is the number of residues and second is the vocabulary size; L is the product of these rather than the first dimension).

The additional analysis of the in silico scores is valuable in assessing the role of in silico screening. The authors provided extensive commentary on their findings in the reviewer responses, some of which could be considered for inclusion in the body of the results section as it appears to me to be interesting (in particular the relative merits of the original function score, the single sub-function scores, and the revised combined hydrophobicity + MinD binding score). To better interpret the discriminative accuracy of these scores it might help to report AUC values or similar, in addition to presenting the scores visually (this might help understand for example the sense in which the revised combined score is 'better'/more significant than the hydrophobicity score alone, which is hard to discern from the bar chart). It would also be relevant to add the sequence identity to wt and to nearest blast hit to the extended figure 7 bar chart to demonstrate that these scores are more effective than naive homologue matching in a direct head-to-head comparison (extended figure 8 hints at this but the comparison with the actual

experimental results is missing, so it is not completely clear how to interpret the differences between the function score and the sequence identity scores). VAE likelihoods would also be of interest as additional scoring functions here, as previous works have shown that the likelihoods of VAEs and other unsupervised generative models are strong zero-shot predictors of protein function [1, 2, 3, 5]. It would be interesting evidence of the effectiveness of the authors' in silico pipeline if it could be demonstrated that their structure-based scores are more effective than sequence likelihoods, or can be used to enhance their predictive power.

Finally, other reviewers comment on the extent to which the recapitulation of function by a designed protein with clear similarities in core residues to known functional proteins is surprising. These are relevant concerns which the authors have sought to address via their emphasis on assessing the effectiveness of their screening pipeline, and by noting that success is by no means guaranteed by simply recapitulating these residues. I am minded to agree with the authors' claim that recapitulation of function with generative methods while incorporating significant numbers of amino acid differences to any known protein is non-trivial and noteworthy. The extent of function-preserving deviation from any known protein reported in this paper is broadly in line with the standards set by other published works using evolution-based generative models [2,3,4,5] (and the function targeted here is more complex, which may or may not be important for evaluating significance). It is nonetheless important to address the concerns about how surprising such results are as far as possible. An additional quantitative analysis that could demonstrate that their scoring pipeline is more effective than simply measuring sequence conservation would be to demonstrate that scores produced by a software such as HMMer (which measures the extent to which a sequence recapitulates the HMM 'profile' of sequences in a given family) are less predictive of function than the scores used in the paper, e.g. by adding HMMer scores to extended figure 7.

References:

1. Riesselman, A. J., Ingraham, J. B., & Marks, D. S. (2018). Deep generative models of genetic variation capture the effects of mutations. *Nature methods*, 15(10), 816–822. <https://doi.org/10.1038/s41592-018-0138-4>
2. Hawkins-Hooker A, Depardieu F, Baur S, Couairon G, Chen A, et al. (2021) Generating functional protein variants with variational autoencoders. *PLOS Computational Biology* 17(2): e1008736. <https://doi.org/10.1371/journal.pcbi.1008736>
3. William P. Russ et al. ,An evolution-based model for designing chorismate mutase enzymes. *Science*369,440-445(2020).DOI:10.1126/science.aba3304
4. Repecka, D., Jauniskis, V., Karpus, L. et al. Expanding functional protein sequence spaces using generative adversarial networks. *Nat Mach Intell* 3, 324–333 (2021). <https://doi.org/10.1038/s42256-021-00310-5>
5. Madani, A., Krause, B., Greene, E.R. et al. Large language models generate functional protein sequences across diverse families. *Nat Biotechnol* 41, 1099–1106 (2023).

<https://doi.org/10.1038/s41587-022-01618-2>

Reviewer #3 (Remarks to the Author):

The basic goal of the paper is to generate MinE variants using generative methods, reduce the number of candidates by picking diverse members. These were screened initially using AlphaFold for interaction with MinD and dimerization and then employing calculations for membrane binding and solubility. The best and worst scoring candidates were screened in vitro for wave formation and then tested in vivo for complementation. Finally, mutants were purified and tested for stimulation of MinD ATPase and oligomerization. From this they end up with one sequence that performs as well as WT MinE but has less than 50% identity.

Not necessary for this paper but I find the v5 mutant interesting. This mutant is not elongated and makes minicells. However, it appears to oscillate (Ext data Fig. 9) with the same frequency as the WT or V25. So why does this result in minicell formation? I don't see this mutant in Fig. 4d. Any explanation for this behavior since it must bind the membrane and stimulate MinD's ATPase activity. It seems to me that if it oscillates at normal frequency in the presence of MinC it should not make minicells. I see that it could not be purified but the fact that it produces normal oscillation indicates it is present and functional, however, it is curious.

Also, v26 also oscillates with near normal frequency but cells are not WT and minicells are produced.

Extended data Fig. 6b. It is not clear to me what the number above the bars refers to (n). They vary from 72 to 124 but I don't see what they correlate with. Please explain in legend or remove.

Reviewer #4 (Remarks to the Author):

The authors addressed all my questions and concerns. I think the article is in a very good shape.

Reviewer #4 (Remarks on code availability):

I'm not qualified to review the code. I have experience with the experimental part of the system described in the paper, not the ML code.

General Remarks

We thank all the reviewers and the editor for their positive assessments of our revised manuscript. To address the remaining concerns, especially the feedback by reviewer #1, we added two more post-hoc analyses of the *in silico* screening and changed Extended Data Fig. 7 to be more visually appealing and clear. Again, we answer the reviewers' comments point by point. We believe that these minor adaptations further improve the clarity of the manuscript and thank the reviewers for the valuable feedback. We hope the reviewers agree with us.

Point by Point Responses

REVIEWER COMMENTS

Reviewer #1 (Remarks to the Author):

The authors have revised the presentation of their paper to clarify its contributions and performed additional analysis that supports the claims made for those contributions. In particular, the authors demonstrate in post-hoc analysis that the *in silico* and *in vitro* screening successfully identify the functional *in vivo* design as the leading candidate after the first two screening stages, providing strongly suggestive evidence that these screening methods are effective and complementary. Additional quantitative analysis further supports these claims to the extent that can be expected for this kind of proof of principle study. My previous concerns have been addressed carefully, the manuscript is improved and seems to me a valuable contribution to the field.

We thank the reviewer for the positive assessment of the improvements we made.

I have minor remaining feedback relating to the revisions. In particular some small pieces of further *in silico* analysis relating to *in silico* scoring functions could strengthen the conclusions further.

The authors could support their choice of an evolution-based generative model more strongly rather than entirely in negative terms (as a fallback due to the difficulty of creating a *de novo* design method). Evolution-based models such as the MSA VAE have had notable success in functional protein design and can be seen as a way of generating diversity while implicitly conditioning on function. Some of this benefit is lost in the current presentation.

We rephrased parts of the Introduction and Results to more positively motivate our choice of generative model (p. 3 l.16-21, p. 4 l. 22-24)

The authors clearly and openly address their non-standard VAE modelling choices in response to my previous comment. This should be helpful to readers unfamiliar with the model class. I found one sentence in the revised description somewhat ambiguous: 'L is the length of the one hot encoded sequence'. It would be easy to interpret this sentence as being equivalent to saying that L is the number of residues, whereas in fact the authors mean that L is the number of binary variables after one-hot encoding the sequence and then flattening it (one-hot encoding produces a matrix whose first dimension is the number of residues and second is the vocabulary size; L is the product of these rather than the first dimension).

We changed the indicated sentence to hopefully make it unambiguous.

The additional analysis of the *in silico* scores is valuable in assessing the role of *in silico* screening. The authors provided extensive commentary on their findings in the reviewer responses, some of which could be considered for inclusion in the body of the results section as it appears to me to be interesting (in particular the relative merits of the original function score, the single sub-function scores, and the revised combined hydrophobicity + MinD binding score). To better interpret the discriminative accuracy of these scores it might help to report AUC values or similar, in addition to presenting the scores visually (this might help understand for example the sense in which the revised combined score is 'better'/more significant than the hydrophobicity score alone, which is hard to discern from the bar chart). It would also be relevant to add the sequence identity to wt and to nearest blast hit to the extended figure 7 bar chart to demonstrate that these scores are more effective than

naive homologue matching in a direct head-to-head comparison (extended figure 8 hints at this but the comparison with the actual experimental results is missing, so it is not completely clear how to interpret the differences between the function score and the sequence identity scores). VAE likelihoods would also be of interest as additional scoring functions here, as previous works have shown that the likelihoods of VAEs and other unsupervised generative models are strong zero-shot predictors of protein function [1, 2, 3, 5]. It would be interesting evidence of the effectiveness of the authors' in silico pipeline if it could be demonstrated that their structure-based scores are more effective than sequence likelihoods, or can be used to enhance their predictive power.

We changed Extended Data Fig. 7 and performed the requested additional analysis. Extended Data Fig. 7 now shows ROC curves and AUC values for all scores, as well as two new scores: (1) scoring based on the ELBO loss generated by averaging over 200 forward passes per synMinE sequence through the VAE and loss calculation, which is the same method as reference [2] mentioned by the reviewer used, and (2) scoring based on HMMER scores when running `hmmsearch` against the MinE hmm profile. In fact, the improved combined score (hydrophobicity + MinD) outperforms also these two scoring methods. We hope the reviewer agrees that the updated visualization is more appealing and that the additional scores further indicate that the “divide-and-conquer” approach is promising. Inspired by the reviewer’s feedback we also include more information on the post-hoc analysis in the main body (p. 7 l. 18-32, p. 11 l. 37 – p. 12 l. 2)

[It should be mentioned that we had a typo in the code for plotting the barchart of the last submission version, which led to a wrong order in the display of sorting by the improved score. This is why the order is different in the updated version (this was only for plotting and did not affect the statistical analysis).]

Finally, other reviewers comment on the extent to which the recapitulation of function by a designed protein with clear similarities in core residues to known functional proteins is surprising. These are relevant concerns which the authors have sought to address via their emphasis on assessing the effectiveness of their screening pipeline, and by noting that success is by no means guaranteed by simply recapitulating these residues. I am minded to agree with the authors' claim that recapitulation of function with generative methods while incorporating significant numbers of amino acid differences to any known protein is non-trivial and noteworthy. The extent of function-preserving deviation from any known protein reported in this paper is broadly in line with the standards set by other published works using evolution-based generative models [2,3,4,5] (and the function targeted here is more complex, which may or may not be important for evaluating significance). It is nonetheless important to address the concerns about how surprising such results are as far as possible. An additional quantitative analysis that could demonstrate that their scoring pipeline is more effective than simply measuring sequence conservation would be to demonstrate that scores produced by a software such as HMMer (which measures the extent to which a sequence recapitulates the HMM 'profile' of sequences in a given family) are less predictive of function than the scores used in the paper, e.g. by adding HMMer scores to extended figure 7.

As mentioned above, we provide HMMER scores in the updated Extended Data Fig. 7.

References:

1. Riesselman, A. J., Ingraham, J. B., & Marks, D. S. (2018). Deep generative models of genetic variation capture the effects of mutations. *Nature methods*, 15(10), 816–822. <https://doi.org/10.1038/s41592-018-0138-4>
2. Hawkins-Hooker A, Depardieu F, Baur S, Couairon G, Chen A, et al. (2021) Generating functional protein variants with variational autoencoders. *PLOS Computational Biology* 17(2): e1008736. <https://doi.org/10.1371/journal.pcbi.1008736>
3. William P. Russ et al. ,An evolution-based model for designing chorismate mutase enzymes.*Science*369,440-445(2020).DOI:10.1126/science.aba3304

4. Repecka, D., Jauniskis, V., Karpus, L. et al. Expanding functional protein sequence spaces using generative adversarial networks. *Nat Mach Intell* 3, 324–333 (2021). <https://doi.org/10.1038/s42256-021-00310-5>

5. Madani, A., Krause, B., Greene, E.R. et al. Large language models generate functional protein sequences across diverse families. *Nat Biotechnol* 41, 1099–1106 (2023). <https://doi.org/10.1038/s41587-022-01618-2>

Reviewer #3 (Remarks to the Author):

The basic goal of the paper is to generate MinE variants using generative methods, reduce the number of candidates by picking diverse members. These were screened initially using Alphafold for interaction with MinD and dimerization and then employing calculations for membrane binding and solubility. The best and worst scoring candidates were screened in vitro for wave formation and then tested in vivo for complementation. Finally, mutants were purified and tested for stimulation of MinD ATPase and oligomerization. From this they end up with one sequence that performs as well as WT MinE but has less than 50% identity.

Not necessary for this paper but I find the v5 mutant interesting. This mutant is not elongated and makes minicells. However, it appears to oscillate (Ext data Fig. 9) with the same frequency as the WT or V25. So why does this result in minicell formation? I don't see this mutant in Fig. 4d. Any explanation for this behavior since it must bind the membrane and stimulate MinD's ATPase activity. It seems to me that if it oscillates at normal frequency in the presence of MinC it should not make minicells. I see that it could not be purified but the fact that it produces normal oscillation indicates it is present and functional, however, it is curious.

Also, v26 also oscillates with near normal frequency but cells are not WT and minicells are produced.

We thank the reviewer for their insightful comments. Indeed, in the screening process, we found some synMinE variants that failed to restore the normal cell phenotype or size distribution compared to the wild type found as minicells or filamentous cells while being able to induce Min oscillations. Essentially, such phenotypes are caused by the lack of proper FtsZ-ring placement. In minicell phenotypes, the effective depolymerization of FtsZ by MinC does not function, allowing the FtsZ-ring assembly even at the cell poles, which produces minicells. In filamentous cells, MinC inhibits FtsZ polymerization everywhere, even in the mid-cell region, causing the elongated cells due to the lack of an FtsZ-ring. Therefore, the abnormal phenotypes are likely caused by less/over-effective MinC colocalization with MinD leading to failing FtsZ-ring placement rather than by MinDE oscillation properties.

However, most importantly, the MinC- and MinE- binding sites on the MinD surface are overlapping and, therefore, competitive MinD binding between MinC and MinE occurs^{1,2}. This leads to an assumption that if the binding affinity between MinD and synMinE is overly strong compared to the wild type, such a variant excludes MinC-MinD interaction, and, hence, MinC cannot regulate FtsZ assembly on the membrane, causing a minicell production (The related discussion has been mentioned on p.9, l. 18-30). Since we have only focused on MinD-MinE and MinE-MinE interactions in this study, and to be fair, such competitive binding between three or more proteins like MinD-MinC vs. MinD-MinE have not been well studied in the protein design field, it is not a surprise that we found synMinE variants that induce abnormal phenotypes. In the same vein, the biochemical characterization of MinD-MinC-MinE competitive interaction might assist that hypothesis, although this was out of our focus in this study. However, this is a fascinating feature to be considered in future protein design studies, because there must be many cellular systems controlling their functions by such competitive interactions between proteins, protein-DNA/RNA, or protein-membrane.

¹Lackner et al., 2003, *J. Bacteriol.*

²Hu et al., 2003, *J. Bacteriol.*

Extended data Fig. 6b. It is not clear to me what the number above the bars refers to (n). They vary from 72 to 124 but I don't see what they correlate with. Please explain in legend or remove.

Here, the (n) indicates the total number of analyzed droplets to calculate the percentage of droplets that contain Min oscillations. The explanation has been added to the legend.

Reviewer #4 (Remarks to the Author):

The authors addressed all my questions and concerns. I think the article is in a very good shape.

Reviewer #4 (Remarks on code availability):

I'm not qualified to review the code. I have experience with the experimental part of the system described in the paper, not the ML code.

We are grateful for the reviewer's positive assessment

REVIEWERS' COMMENTS

Reviewer #1 (Remarks to the Author):

The authors again carefully addressed my concerns, adding further post hoc analysis of the in silico screening stage that provides further evidence that the proposed structure-based scores are informative of function. The comparison with sequence-based scores is interesting: while sequence-based scores are effective (indeed more effective than the original combined structure-based score) the best-performing post-hoc selected structure-based scores appear to be adding extra and potentially complementary information, offering support for the authors' claims regarding the effectiveness of their proposed in silico strategy, and suggesting that scoring approaches that combine the two sources of information might be a fruitful avenue in future work.

I have no further concerns, and believe the manuscript to be an interesting contribution to the field of protein design.